

**Analysis of Groundwater Response to Oscillatory Pumping Test in**
**Unconfined Aquifers: Consider the Effects of Initial Condition and**
**Wellbore Storage**
**Ching-Sheng Huang[a], Ya-Hsin Tsai[b], Hund-Der Yeh[b*] and Tao Yang[a*]**
[a] State Key Laboratory of Hydrology-Water Resources and Hydraulic Engineering, Center for
Global Change and Water Cycle, Hohai University, Nanjing 210098, China
[b] Institute of Environmental Engineering, National Chiao Tung University, Hsinchu 300,
Taiwan
[*] Corresponding authors:
Hund-Der Yeh; E-mail: hdyeh@mail.nctu.edu.tw; Tel.: +886-3-5731910; fax: +886-3-

11   5725958

Tao Yang; E-mail: tao.yang@hhu.edu.cn; Tel.: +86-13770918075
Submission to *Hydrology and Earth System Sciences* on 12 April 2018
**Key points**
1.  An analytical solution of the hydraulic head due to oscillatory pumping test in unconfined

18       aquifers is presented.

2.  The effects of wellbore storage and initial condition of static groundwater before the test

20       are analyzed.

3.  The present solution agrees well to head fluctuation data taken from a field oscillatory

22       pumping test.





## Abstract

Oscillatory pumping test (OPT) is an alternative to constant-head and constant-rate pumping
tests for determining aquifer hydraulic parameters without water extraction. There is a large
number of analytical models presented for the analyses of OPT. The combined effects of
wellbore storage and initial condition regarding the hydraulic head prior to OPT are commonly
neglected in the existing models. This study aims to develop a new model for describing the
hydraulic head fluctuation induced by OPT in an unconfined aquifer. The model contains a
typical flow equation with an initial condition of static water table, inner boundary condition
specified at the rim of a finite-radius well for incorporating wellbore storage effect, and
linearized free surface equation describing water table movement. The analytical solution of
the model is derived by the Laplace transform and finite integral transform. Sensitivity analysis
is carried out for exploring head response to the change in each of hydraulic parameters. Results
suggest that head fluctuation due to OPT starts from the initial condition and gradually tends
to simple harmonic motion (SHM) after a certain pumping time. A criterion for estimating the
time to have SHM since OPT is graphically presented. The validity of assuming an
infinitesimal well radius without wellbore storage effect is investigated. The present solution
agrees well to head fluctuation data observed at the Boise hydrogeophysical research site in
southwestern Idaho.
**KEYWORDS:** oscillatory pumping test, analytical solution, free surface equation, initial
condition, wellbore storage






# NOTATION

| | |
|---|---|
| $a$ | $\sigma/\mu$ |
| $b$ | Aquifer thickness |
| $\bar{b}$ | Dimensionless aquifer thickness, i.e., $\bar{b} = b/r_w$ |
| $h$ | Hydraulic head |
| $\bar{h}$ | Dimensionless Hydraulic head, i.e., $\bar{h} = (2\pi b K_r h)/|Q|$ |
| $K_r, K_z$ | Aquifer horizontal and vertical hydraulic conductivities, respectively |
| $P$ | Period of oscillatory pumping rate |
| $p$ | Laplace parameter |
| $Q$ | Amplitude of oscillatory pumping rate |
| $R$ | Radius of influence |
| $\bar{R}$ | Dimensionless radius of influence, i.e., $\bar{R} = R/r_w$ |
| $r$ | Radial distance from the center of pumping well |
| $\bar{r}$ | Dimensionless radial distance, i.e., $\bar{r} = r/r_w$ |
| $r_c$ | Outer radius of pumping well |
| $r_w$ | Inner radius of pumping well |
| $S_s, S_y$ | Specific storage and specific yield, respectively |
| $t$ | Time since pumping |
| $\bar{t}$ | Dimensionless pumping time, i.e., $\bar{t} = (K_r\,t)/(S_s\,r_w^2)$ |
| $z$ | Elevation from aquifer bottom |
| $\bar{z}$ | Dimensionless elevation, i.e., $\bar{z} = z/b$ |
| $\alpha$ | $r_c^2/(2r_w^2 S_s b)$ |
| $\beta_n, \beta_m$ | Roots of Eqs. (19) and (36), respectively |
| $\gamma$ | $S_s\,r_w^2\,\omega/K_r$ |
| $\kappa$ | $K_z/K_r$ |
| $\mu$ | $\kappa/\bar{b}^2$ |
| $\sigma$ | $S_y/(S_s\,b)$ |
| $\omega$ | Frequency of oscillatory pumping rate, i.e., $\omega = 2\pi/P$ |




## 1. Introduction


Numerous attempts have been made by researchers to the study of oscillatory pumping test
(OPT) that is an alternative to constant-rate and constant-head pumping tests for determining
aquifer hydraulic parameters (e.g., Vine et al., 2016; Christensen et al., 2017; Watlet et al.,
2018). The concept of OPT was first proposed by Kuo (1972) in the petroleum literature. The
process of OPT contains extraction stages and injection stages. The pumping rate, in other
words, varies periodically as a sinusoidal function of time. Compared with traditional constant-
rate pumping, OPT in contaminated aquifers has the following advantages: (1) low cost because
of no disposing contaminated water from the well, (2) reduced risk of treating contaminated
fluid, (3) smaller contaminant movement, and (4) stable signal easily distinguished from
background disturbance such as tide effect and varying river stage (e.g., Spane and Mackley,
2011). However, OPT has the disadvantages including the need of an advanced apparatus
producing periodic rate and the problem of signal attenuation in remote distance from the
pumping well. Oscillatory hydraulic tomography adopts several oscillatory pumping wells with
different frequencies (e.g., Yeh and Liu, 2000; Cardiff et al., 2013; Zhou et al., 2016;
Muthuwatta, et al., 2017). Aquifer heterogeneity can be mapped by analyzing multiple data
collected from observation wells. Cardiff and Barrash (2011) reviewed articles associated with
hydraulic tomography and classified them according to nine categories in a table.

Various groups of researchers have worked with analytical and numerical models for OPT;

each group has its own model and investigation. For example, Black and Kipp (1981) assumed
the response of confined flow to OPT as simple harmonic motion (SHM) in the absence of an
initial condition. Cardiff and Barrash (2014) built an optimization formulation strategy using
the Black and Kipp analytical solution. Dagan and Rabinovich (2014) also assumed hydraulic
head fluctuation as SHM for OPT at a partially penetrating well in unconfined aquifers. Cardiff
et al. (2013) characterized aquifer heterogeneity using the finite element-based COMSOL
software that adopts SHM hydraulic head variation for OPT. On the other hand, Rasmussen et



al. (2003) found that hydraulic head response tends to SHM after a certain period of pumping

time when considering an initial condition prior to OPT. Bakhos et al. (2014) used the

Rasmussen et al. (2003) analytical solution to quantify the time after which hydraulic head

fluctuation can be regarded as SHM since OPT began. As shown above, existing models for

OPT have either assumed hydraulic head fluctuation as SHM without an initial condition or

ignored the effect of wellbore storage with considering an infinitesimal well radius.

Field applications of OPT for determining aquifer parameters have been conducted in

recent years. Rasmussen et al. (2003) estimated aquifer hydraulic parameters based on $1-2.5$-

hour period of OPT at the Savannah River site. Maineult et al. (2008) observed spontaneous

potential temporal variation in aquifer diffusivity at a study site in Bochum, Germany. Fokker

et al. (2012; 2013) presented spatial distributions of aquifer transmission and storage

coefficient derived from curve fitting based on a numerical model and field data from

experiments at the southern city-limits of Bochum, Germany. Rabinovich et al. (2015)

estimated aquifer parameters of equivalent hydraulic conductivity, specific storage and specific

yield at the Boise Hydrogeophysical Research Site (BHRS) by curve fitting based on

observation data and the Dagan and Rabinovich analytical solution. They conclude that the

equivalent hydraulic parameters can represent the actual aquifer heterogeneity of the study site.

Although a large number of studies have been made on development of analytical models

for OPT, little is known about the combined effects of wellbore storage and initial condition

prior to OPT. Analytical solution to such a question will not only have important physical

implications but also shed light on OPT model development. This study builds an improved

model describing hydraulic head fluctuation induced by OPT in an unconfined aquifer. The

model is composed of a typical flow equation with the initial condition of static water table, an

inner boundary condition specified at the rim of the pumping well for incorporating wellbore

storage effect, and a first-order free surface equation describing the movement of aquifer water

table. The analytical solution of the model is derived by the methods of Laplace transform and

finite integral transform. Based on the present solution, sensitivity analysis is performed to



explore the hydraulic head in response to the change in each of hydraulic parameters. The
quantitative criteria for excluding the individual effects of wellbore storage and the initial
condition are discussed. The radius of influence induced by OPT is investigated for engineering
applications. In addition, curve fitting of the present solution to head fluctuation data recorded
at BHRS is presented.
**2. Methodology**
**2.1. Mathematical model**
Consider an oscillatory pumping at a fully penetrating well in an unconfined aquifer illustrated
in Fig. 1. The aquifer is of unbound lateral extent with a finite thickness $b$. The radial distance
from the centerline of the well is $r$; an elevation from the impermeable bottom of the aquifer is
$z$. The well has inner radius $r_c$ and outer radius $r_w$.

The flow equation describing spatiotemporal head distribution in aquifers can be written

as:
$K_r \left( \frac{\partial^2 h}{\partial r^2} + \frac{1}{r} \frac{\partial h}{\partial r} \right) + K_z \frac{\partial^2 h}{\partial z^2} = S_s \frac{\partial h}{\partial t}$   for $r_w \leq r < \infty$, $0 \leq z \leq b$ and $t \geq 0$          (1)
where $h(r, z, t)$ is hydraulic head at location ($r$; $z$) and time $t$; $K_r$ and $K_z$ are respectively
the radial and vertical hydraulic conductivities; $S_s$ is the specific storage. Consider water table
as a reference datum where the elevation head is set to zero; the initial condition is expressed
as:
$h = 0$ at $t = 0$                                                                (2)
The rim of the wellbore is regarded as an inner boundary, which provides the associated
condition as:
$2\pi r_w K_r b \frac{\partial h}{\partial r} = Q \sin(\omega t) + \pi r_c^2 \frac{\partial h}{\partial t}$ at $r = r_w$                          (3)
where $Q$ and $\omega$ are respectively the amplitude and frequency of oscillatory pumping rate; is
frequency. The first term on the right-hand side (RHS) of Eq. (3) represents an oscillatory
pumping rate, and the second term represents the volume change within the well reflecting
wellbore storage effect. Water table movement can be defined by the first-order free surface





equation proposed by Neuman (1972) as
$K_z \frac{\partial h}{\partial z} = -S_y \frac{\partial h}{\partial t}$   at   $z = b$           (4)
where $S_y$ is the specific yield. The impervious aquifer bottom is under the no-flow condition:
$\frac{\partial h}{\partial z} = 0$   at   $z = 0$           (5)
The hydraulic head far away from the well remains constant and is expressed as
$\lim_{r \to \infty} h(r, z, t) = 0$           (6)

Define dimensionless variables and parameters as follows:

$\bar{h} = \frac{2 \pi b K_r}{Q} h$,   $\bar{r} = \frac{r}{r_w}$,   $\bar{z} = \frac{z}{b}$,   $\bar{t} = \frac{K_r}{S_s r_w^2} t$,   $\bar{b} = \frac{b}{r_w}$,
$\alpha = \frac{r_c^2}{2 r_w^2 S_s b}$,   $\gamma = \frac{S_s r_w^2}{K_r} \omega$,   $\kappa = \frac{K_z}{K_r}$,   $\mu = \frac{\kappa}{\bar{b}^2}$,   $\sigma = \frac{S_y}{S_s b}$,   $a = \frac{\sigma}{\mu}$      (7)
where the overbar stands for a dimensionless symbol. Note that the magnitude of $\alpha$ dominates
wellbore storage effect (Papadopulos and Cooper, 1967) and $\gamma$ is a dimensionless frequency
parameter. With Eq. (7), the dimensionless forms of Eqs. (1) - (6) become, respectively,
$\frac{\partial^2 \bar{h}}{\partial \bar{r}^2} + \frac{1}{\bar{r}} \frac{\partial \bar{h}}{\partial \bar{r}} + \mu \frac{\partial^2 \bar{h}}{\partial \bar{z}^2} = \frac{\partial \bar{h}}{\partial \bar{t}}$   for   $1 \le \bar{r} < \infty$, $0 \le \bar{z} < 1$ and $\bar{t} \ge 0$     (8)
$\bar{h} = 0$ at $\bar{t} = 0$           (9)
$\frac{\partial \bar{h}}{\partial \bar{r}} = \sin(\gamma \bar{t}) + \alpha \frac{\partial \bar{h}}{\partial \bar{t}}$ at $\bar{r} = 1$           (10)
$\frac{\partial \bar{h}}{\partial \bar{z}} = -a \frac{\partial \bar{h}}{\partial \bar{t}}$ at $\bar{z} = 1$           (11)
$\frac{\partial \bar{h}}{\partial \bar{z}} = 0$ at $\bar{z} = 0$           (12)
$\lim_{\bar{r} \to \infty} \bar{h}(\bar{r}, \bar{z}, \bar{t}) = 0$           (13)
The transient solution of the dimensionless head $\bar{h}$ satisfies Eqs. (8) - (13) with the initial
condition Eq. (9). Here we define a pseudo-steady state solution $\bar{h}_s$ to the model of Eqs. (8)
and (10) - (13) with $\sin(\gamma \bar{t})$ in Eq. (10) replaced by $\text{Im}(e^{i\gamma \bar{t}})$, Im(-) being the imaginary
part of a complex number, and $i$ being the imaginary unit. The pseudo-steady state model
accounts for SHM of head fluctuation after a certain period of pumping time.





**2.2. Transient solution for unconfined aquifer**
The Laplace transform and finite integral transform are applied to solve Eqs. (8) - (13) (Liang
et al., 2017). The former converts $\bar{h}(\bar{r}, \bar{z}, \bar{t})$ into $\hat{h}(\bar{r}, \bar{z}, p)$, $\partial\bar{h}/\partial\bar{t}$ in Eq. (8), (10) and (11)
into $p\hat{h}$, and $\sin(\gamma\bar{t})$ in Eq. (10) into $\gamma/(p^2 + \gamma^2)$ with the Laplace parameter $p$. The result
of Eq. (8) in the Laplace domain can be written as
$$\frac{\partial^2\hat{h}}{\partial\bar{r}^2} + \frac{1}{\bar{r}}\frac{\partial\hat{h}}{\partial\bar{r}} + \mu\frac{\partial^2\hat{h}}{\partial\bar{z}^2} = p\hat{h} \tag{14}$$

The transformed boundary conditions in $r$ and $z$ directions are expressed as
$$\frac{\partial\hat{h}}{\partial\bar{r}} = \frac{\gamma}{p^2 + \gamma^2} + \alpha p\hat{h} \text{ at } \bar{r} = 1 \tag{15}$$

$$\frac{\partial\hat{h}}{\partial\bar{z}} = -ap\hat{h} \text{ at } \bar{z} = 1 \tag{16}$$

$$\frac{\partial\hat{h}}{\partial\bar{z}} = 0 \text{ at } \bar{z} = 0 \tag{17}$$

$$\lim_{\bar{r}\to\infty} \hat{h}(\bar{r}, \bar{z}, p) = 0 \tag{18}$$

The finite integral transform proposed by Latinopoulos (1985) is applied to Eqs. (14) -

(17). The definition of the transform is given in Appendix A. Using the property of the
transform converts $\hat{h}(\bar{r}, \bar{z}, p)$ into $\tilde{h}(\bar{r}, \beta_n, p)$, $\mu\,\partial^2\hat{h}/\partial\bar{z}^2$ in Eq. (14) into $-\mu\beta_n^2\tilde{h}$, and
$\gamma/(p^2 + \gamma^2)$ in Eq. (15) into $\gamma F_t \sin\beta_n /(p^2 + \gamma^2)$ where $n \in (1,2,3,\dots\infty)$ ; $F_t =$
$\sqrt{2(\beta_n^2 + a^2 p^2)/(\beta_n^2 + a^2 p^2 + ap)}$; $\beta_n$ is the positive roots of the equation:
$$\tan\beta_n = ap/\beta_n \tag{19}$$
The method to find the roots of $\beta_n$ is discussed in section 2.3. Eq. (14) then becomes an
ordinary differential equation (ODE) denoted as
$$\frac{\partial^2\tilde{h}}{\partial\bar{r}^2} + \frac{1}{\bar{r}}\frac{\partial\tilde{h}}{\partial\bar{r}} - \mu\beta_n^2\tilde{h} = p\tilde{h} \tag{20}$$

with the transformed Eqs. (18) and (15) written, respectively, as
$$\lim_{\bar{r}\to\infty} \tilde{h}(\bar{r}, \beta_n, p) = 0 \tag{21a}$$

$$\frac{\partial\tilde{h}}{\partial\bar{r}} = \frac{\gamma F_t \sin\beta_n}{\beta_n(p^2 + \gamma^2)} + \alpha p\tilde{h} \text{ at } \bar{r} = 1 \tag{21b}$$

Note that the transformation from Eq. (14) to (20) is applicable only for the no-flow condition



specified at $\bar{z} = 0$ (i.e., Eq. (17)) and third-type condition specified at $\bar{z} = 1$ (i.e., Eq. (16)).
Solve Eq. (20) with (21a) and (21b), and we obtain:
$\tilde{h}(\bar{r}, \beta_n, p) = -\frac{\gamma F_t \sin \beta_n K_0(r\lambda)}{\beta_n(p^2+\gamma^2)(p\alpha K_0(\lambda)+\lambda K_1(\lambda))}$ (22)
with
$\lambda = \sqrt{p + \mu \beta_n^2}$ (23)
where $K_0(-)$ and $K_1(-)$ is the modified Bessel function of the second kind of order zero
and one, respectively. Applying the inverse Laplace transform and inverse finite integral
transform to Eq. (22) results in the transient solution expressed as
$\bar{h}(\bar{r}, \bar{z}, \bar{t}) = \bar{h}_{\exp}(\bar{r}, \bar{z}, \bar{t}) + \bar{h}_{\mathrm{SHM}}(\bar{r}, \bar{z}, \bar{t})$ (24a)
with
$\bar{h}_{\exp}(\bar{r}, \bar{z}, \bar{t}) = \frac{-2}{\pi} \sum_{n=1}^{\infty} \int_0^{\infty} \cos(\beta_n \bar{z}) \, \mathrm{Im}(\gamma \varepsilon_1 \varepsilon_2 \exp(p_0 \bar{t})) \, d\zeta$ (24b)
$\bar{h}_{\mathrm{SHM}}(\bar{r}, \bar{z}, \bar{t}) = \bar{A}_t(\bar{r}, \bar{z}) \cos(\gamma \, \bar{t} - \phi_t(\bar{r}, \bar{z}))$ (24c)
$\bar{A}_t(\bar{r}, \bar{z}) = \sqrt{a_t(\bar{r}, \bar{z})^2 + b_t(\bar{r}, \bar{z})^2}$ (24d)
$a_t(\bar{r}, \bar{z}) = \frac{2}{\pi} \sum_{n=1}^{\infty} \int_0^{\infty} \cos(\beta_n \bar{z}) \, \mathrm{Im}(\varepsilon_1 \varepsilon_2 \, p_0) \, d\zeta$ (24e)
$b_t(\bar{r}, \bar{z}) = \frac{2\gamma}{\pi} \sum_{n=1}^{\infty} \int_0^{\infty} \cos(\beta_n \bar{z}) \, \mathrm{Im}(\varepsilon_1 \varepsilon_2) \, d\zeta$ (24f)
$\phi_t(\bar{r}, \bar{z}) = \cos^{-1}(b_t(\bar{r}, \bar{z})/\bar{A}_t(r, \bar{z}))$ (24g)
$\varepsilon_1 = \sin \beta_n \, K_0(\bar{r}\lambda_0)/\left(\beta_n(p_0^2 + \gamma^2)(p_0 \alpha K_0(\lambda_0) + \lambda_0 K_1(\lambda_0))\right)$ (24h)
$\varepsilon_2 = (\beta_n^2 + a^2 p_0^2)/(\beta_n^2 + a^2 p_0^2 + a p_0)$ (24i)
$p_0 = -\zeta - \mu \beta_n^2$ (24j)
$\lambda_0 = \sqrt{\zeta} i$ (24k)
The detailed derivation of Eqs. (24a) – (24k) is presented in Appendix B. The first RHS term
in Eq. (24a) due to the initial condition exhibits exponential decay since pumping began; the
second term defines SHM with amplitude $\bar{A}_t(\bar{r}, \bar{z})$ and phase shift $\phi_t(\bar{r}, \bar{z})$ at a given point
$(\bar{r}, \bar{z})$. The numerical results of the integrals in Eqs. (24b), (24e) and (24f) are obtained by the
Mathematica NIntegrate function.





**2.3. Calculation of $\beta_n$**
The eigenvalues $\beta_1,\ldots,\ \beta_n$, the roots of Eq. (19) with $p$ replaced by $p_0$ in Eq. (24j), can
be determined by applying the Mathematica function FindRoot based on Newton's method
with reasonable initial guesses. The roots are located at the intersection of the curves plotted
by the RHS and left-hand side (LHS) functions of $\beta_n$ in Eq. (19). The roots are very close to
the vertical asymptotes of the periodical tangent function $\tan\beta_n$. The initial guess for each $\beta_n$
can be considered as $(2n-1)\pi/2 + \delta$ where $n \in (1,2,\ldots\infty)$ and $\delta$ is a small positive
value set to $10^{-10}$ to prevent the denominator in Eq. (19) from zero.
**2.4. Transient solution for confined aquifer**
When $S_y = 0$ (i.e., $\sigma = 0$), Eq. (11) reduces to $\partial\bar{h}/\partial\bar{z} = 0$ for a no-flow condition at the top
of the aquifer, indicating that the unconfined aquifer becomes a confined one. Under this
condition, Eq. (19) becomes $\tan\beta_n = 0$ with roots $\beta_n = 0,\ \pi,\ 2\pi,\ \ldots,\ n\pi,\ \ldots,\ \infty$; Eq. (24i)
reduces to $\varepsilon_2 = 1$; factor 2 in Eqs. (24b), (24e) and (24f) is replaced by unity. The analytical
solution of the transient head for the confined aquifer can be expressed as
$\bar{h}(\bar{r},\bar{t}) = \bar{h}_{\exp}(\bar{r},\bar{t}) + \bar{h}_{\mathrm{SHM}}(\bar{r},\bar{t})$ (25a)
with
$\bar{h}_{\exp}(\bar{r},\bar{t}) = \frac{-1}{\pi}\int_0^\infty \mathrm{Im}(\varepsilon_1\gamma\exp(-\zeta\bar{t}))\,d\zeta$ (25b)
$\bar{h}_{\mathrm{SHM}}(\bar{r},\bar{t}) = \bar{A}_t(\bar{r})\cos(\gamma\bar{t} - \phi_t(\bar{r}))$ (25c)
$\bar{A}_t(\bar{r}) = \sqrt{a_t(\bar{r})^2 + b_t(\bar{r})^2}$ (25d)
$a_t(\bar{r}) = \frac{1}{\pi}\int_0^\infty \mathrm{Im}(-\varepsilon_1\zeta)\,d\zeta$ (25e)
$b_t(\bar{r}) = \frac{\gamma}{\pi}\int_0^\infty \mathrm{Im}(\varepsilon_1)\,d\zeta$ (25f)
$\phi_t(\bar{r}) = \cos^{-1}\big(b_t(\bar{r})/\bar{A}_t(\bar{r})\big)$ (25g)
$\varepsilon_1 = K_0(\bar{r}\lambda_0)/((p_0^2 + \gamma^2)(-\alpha\zeta K_0(\lambda_0) + \lambda_0 K_1(\lambda_0)))$ (25h)
Note that Eq. (24h) reduces to Eq. (25h) based on $\beta_n = 0$ and L' Hospital's rule and gives
$\varepsilon_1 = 0$ for the other roots $\beta_n = \pi,\ 2\pi,\ \ldots,\ n\pi$. This causes that Eqs. (25a) – (25h) are
independent of dimensionless elevation $\bar{z}$, indicating only horizontal flow in the confined



aquifer.

### 2.5. Pseudo-steady state solution for unconfined aquifer

The pseudo-steady state solution $\bar{h}_s$ satisfies the following form (Dagan and Rabinovich,

2014).

$\quad \bar{h}_s(\bar{r}, \bar{z}, \bar{t}) = \text{Im}\big(\bar{H}(\bar{r}, \bar{z}) \, e^{i\gamma\bar{t}}\big)$ $\hspace{3cm}$ (26)
where $\bar{H}(\bar{r}, \bar{z})$ is a space function of $\bar{r}$ and $\bar{z}$. Substituting Eq. (26) and $\partial\bar{h}_s/\partial\bar{t} =$
$\text{Im}\big(i\gamma\bar{H}(\bar{r}, \bar{z}) \, e^{i\gamma\bar{t}}\big)$ into the pseudo-steady state model results in
$\quad \dfrac{\partial^2\bar{H}}{\partial\bar{r}^2} + \dfrac{1}{\bar{r}}\dfrac{\partial\bar{H}}{\partial\bar{r}} + \mu\dfrac{\partial^2\bar{H}}{\partial\bar{z}^2} = i\gamma\bar{H}$ $\hspace{3cm}$ (27)
$\quad \dfrac{\partial\bar{H}}{\partial\bar{r}} = 1 + i\alpha\gamma\bar{H}$ at $\bar{r} = 1$ $\hspace{3cm}$ (28)
$\quad \dfrac{\partial\bar{H}}{\partial\bar{z}} = -i\alpha\gamma\bar{H}$ at $\bar{z} = 1$ $\hspace{3cm}$ (29)
$\quad \dfrac{\partial\bar{H}}{\partial\bar{z}} = 0$ at $\bar{z} = 0$ $\hspace{3cm}$ (30)
$\quad \lim\limits_{\bar{r}\to\infty} \bar{H} = 0$ $\hspace{3cm}$ (31)
Again, taking the finite integral transform to Eqs. (27) - (31) yields
$\quad \dfrac{\partial^2\tilde{H}}{\partial\bar{r}^2} + \dfrac{1}{\bar{r}}\dfrac{\partial\tilde{H}}{\partial\bar{r}} - \mu\beta_m^2\tilde{H} = i\gamma\tilde{H}$ $\hspace{3cm}$ (32)
$\quad \dfrac{\partial\tilde{H}}{\partial\bar{r}} = \dfrac{\sin\beta_m}{\beta_m}F_s + i\alpha\gamma\tilde{H}$ at $\bar{r} = 1$ $\hspace{3cm}$ (33)
$\quad \lim\limits_{\bar{r}\to\infty} \tilde{H} = 0$ $\hspace{3cm}$ (34)
$\quad F_s = \sqrt{2(\beta_m^2 - a^2\gamma^2)/(\beta_m^2 - a^2\gamma^2 + ia\gamma)}$ $\hspace{3cm}$ (35)
where $\beta_m = c_m + d_m i$ is a complex number being the roots of the equation:
$\quad \beta_m \tan\beta_m = ia\gamma$ $\hspace{3cm}$ (36)
The method to determine $\beta_m$ is given in section 2.6. Solving Eq. (32) with (33) and (34)
results in
$\quad \tilde{H}(\bar{r}, \beta_m) = F_s \dfrac{i\sin(\beta_m)K_0(\bar{r}\lambda)}{\beta_m(\alpha\gamma K_0(\lambda) - i\lambda K_1(\lambda))}$ $\hspace{3cm}$ (37)
where $\lambda = \sqrt{\gamma i + \mu\beta_m^2}$. After taking the inverse finite integral transform to Eq. (37) and





applying the formula of $e^{i\gamma\bar{t}} = \cos(\gamma\bar{t}) + i\sin(\gamma\bar{t})$ to the result, the pseudo-steady state
solution can be expressed as
$\bar{h}_s(\bar{r}, \bar{z}, \bar{t}) = \bar{A}_s(\bar{r}, \bar{z}) \cos(\gamma t - \phi_s(\bar{r}, \bar{z}))$      (38a)
with
$\bar{A}_s(\bar{r}, \bar{z}) = \sqrt{a_s(\bar{r}, \bar{z})^2 + b_s(\bar{r}, \bar{z})^2}$      (38b)
$a_s(\bar{r}, \bar{z}) = \mathrm{Re}(\sum_{m=1}^{\infty} D(\bar{r}, \beta_m) \cos(\beta_m \bar{z}))$      (38c)
$b_s(\bar{r}, \bar{z}) = \mathrm{Im}(\sum_{m=1}^{\infty} D(\bar{r}, \beta_m) \cos(\beta_m \bar{z}))$      (38d)
$\phi_s(\bar{r}, \bar{z}) = \cos^{-1}(b_s(\bar{r}, \bar{z})/A_s(\bar{r}, \bar{z}))$      (38e)
$D(\bar{r}, \beta_m) = iF_s^2 \sin\beta_m K_0(\bar{r}\lambda) / (\beta_m(\alpha\gamma K_0(\lambda) - i\lambda K_1(\lambda)))$      (38f)
where Re(-) is the real part of a complex number. Eq. (38a) indicates SHM for the response of
the hydraulic head at any point to oscillatory pumping.
**2.6 Calculation of $\beta_m$**
Substituting   $\beta_m = c_m + d_m i$   and   $\tan\beta_m = \sin(2c_m)/\tau + i\sinh(2d_m)/\tau$   with   $\tau =$
$\cos(2c_m) + \cosh(2d_m)$ into Eq. (36) and separating the real and imaginary parts of the result
leads to the following two equations:
$\sin(2c_m)/\tau = a\gamma d_m/(c_m^2 + d_m^2)$      (39)
and
$\sinh(2d_m)/\tau = a\gamma c_m/(c_m^2 + d_m^2)$      (40)
Noted that Eqs. (39) and (40) are respectively from the real and imaginary parts. The values of
$c_m$ and $d_m$ can be determined by the Mathematica function FindRoot with the initial guesses
of $\pi m/2$ for $c_m$ and $10^{-4}$ for $d_m$.
**2.7 Pseudo-steady state solution for confined aquifers**
Again, when $S_y = 0$ (i.e., $\sigma = 0$ ), Eq. (36) reduces to $\tan\beta_m = 0$ with roots $\beta_m = 0$, $\pi$,
$2\pi$, …, $m\pi$, …, $\infty$; factor 2 in Eq. (35) is replaced by unity. Eq. (38f) then becomes
$D(\bar{r}) = \begin{cases} 0 & \text{for } \beta_m \neq 0 \\ 2iK_0(\bar{r}\lambda)/(\alpha\gamma K_0(\lambda) - i\lambda K_1(\lambda)) & \text{for } \beta_m = 0 \end{cases}$      (41)





which is obtained by applying L' Hospital's rule when $\beta_m = 0$. With Eq. (41), Eqs. (38c) and
(38d) reduces, respectively, to
$\quad a_s(\bar{r}) = \mathrm{Re}\left(\frac{i\,K_0(\bar{r}\lambda)}{\alpha\gamma K_0(\lambda) - i\lambda K_1(\lambda)}\right)$ (42a)
and
$\quad b_s(\bar{r}) = \mathrm{Im}\left(\frac{i\,K_0(\bar{r}\lambda)}{\alpha\gamma K_0(\lambda) - i\lambda K_1(\lambda)}\right)$ (42b)
which are independent of dimensionless elevation $\bar{z}$, indicating horizontal confined flow.
Based on Eqs. (41), (42a) and (42b), the pseudo-steady state solution for confined aquifers can
be expressed as:
$\quad \bar{h}_s(\bar{r}, \bar{t}) = \bar{A}_s(\bar{r}) \cos(\gamma t - \phi_s(\bar{r}))$ (43a)
with
$\quad \bar{A}_s(\bar{r}) = \sqrt{a_s(\bar{r})^2 + b_s(\bar{r})^2}$ (43b)
$\quad \phi_s(\bar{r}) = \cos^{-1}(b_s(\bar{r})/A_s(\bar{r}))$ (43c)
**2.8 Sensitivity analysis**
Sensitivity analysis evaluates hydraulic head variation in response to the change in each of $K_r$,
$K_z$, $S_s$, $S_y$, and $\omega$. The normalized sensitivity coefficient can be defined as (McCuen, 1985)
$\quad S_i = P_i \frac{\partial X}{\partial P_i}$ (44)
where $S_i$ is the sensitivity coefficient of $i$th parameter; $P_i$ is the magnitude of the $i$th input
parameter; $X$ represents the present solution in dimensional form. Eq. (44) can be approximated
as
$\quad S_i = P_i \frac{X(P_i + \Delta P_i) - X(P_i)}{\Delta P_i}$ (45)
where $\Delta P_i$, a small increment, is chosen as $10^{-3} P_i$.
**3. Results and Discussion**
In the following sections, we demonstrate the response of the hydraulic head to oscillatory
pumping using the present solution. The default values in calculation are $b = 20$ m, $Q = 1$ L/s,
$r_c = 0.06$ m, $r_w = 0.05$ m, $K_r = 10^{-4}$ m/s, $K_z = 10^{-5}$ m/s, $S_s = 10^{-5}$ m$^{-1}$, $S_y = 0.1$, $\omega = 2\pi/30$ s$^{-1}$, $r$





$= r_w$ and $z = 10$ m. The corresponding dimensionless parameters are $\alpha = 3600$, $\gamma = 5.24 \times 10^{-5}$,
$\kappa = 0.1$, $\mu = 6.2 \times 10^{-7}$, and $\sigma = 500$. The practical ranges for dimensionless parameters are
$0.1 \leq \kappa \leq 0.5$, $10 \leq \sigma \leq 10^5$, $10^{-1} \leq \alpha \leq 10^5$ and $10^{-6} \leq \gamma \leq 1$.
**3.1. Transient head fluctuation affected by the initial condition**
Figure 2 demonstrates dimensional hydraulic head predicted by the present transient solution
$h = h_{\exp} + h_{\mathrm{SHM}}$ and the pseudo-steady state solution $h_s$ for unconfined aquifers. The head
fluctuation defined by $h$ starts from $h = 0$ at $t = 0$ and approaches SHM that can be
predicted by $h_{\mathrm{SHM}}$ when $h_{\exp} \cong 0$ m after $t = 219$ sec. On the other hand, $h_{\mathrm{SHM}}$ with about
13 sec shift of time predicts very close SHM to the pseudo-steady state solution with error less
than 3%. This example indicates that the present transient solution $h$ can be expressed as $h =$
$h_{\exp} + h_s$ with a certain time shift so that head fluctuation starts from $h = 0$ at $t = 0$.
Define an ignorable dimensionless head change as $|\bar{h}| < 10^{-2}$ (i.e., $|h| < 1$ mm)
according to $\bar{h} = (2\pi b K_r / Q)h$ for the practical ranges of $b K_r \geq 10^3$ m$^2$/d and $Q \leq 10^2$
m$^3$/d (Rasmussen et al. 2003). Define $\bar{t}_s$ as a dimensionless transient time to have
$\bar{h}_{\exp}(\bar{r}, \bar{z}, \bar{t}) = 10^{-2}$ (or $\bar{h} \cong \bar{h}_{\mathrm{SHM}}$). The time can be estimated using the Mathematica
function FindRoot to solve the equation that
$\left| \bar{h}_{\exp}(1, 0.5, \ \bar{t}_s) \right| = 10^{-2}$ (46)
Figure 3 displays the curve of dimensionless frequency $\gamma$ versus the largest predicted $\bar{t}_s$. The
curve is plotted based on the values of $\kappa = 0.1$, $\alpha = 10^5$ and $\sigma = 500$. When $\gamma \leq 2.7 \times 10^{-3}$,
the value of $\bar{t}_s$ decreases with increasing $\gamma$. When $\gamma > 2.7 \times 10^{-3}$, $\bar{t}_s$ can be regarded as
zero because a numerical result from the LHS function of Eq. (46) is smaller than $10^{-2}$ for any
value of $\bar{t}_s$. Note that $\bar{t}_s$ increases with decreasing $\kappa$ so we choose the smallest of the
practical range $0.1 \leq \kappa \leq 0.5$. Variations in dimensionless parameters $\sigma$ and $\alpha$ have
insignificant effect on $\bar{t}_s$ prediction. The largest $\bar{t}_s$ is about $2.45 \times 10^6$ that equals 10 min
obtained by $t_s = S_s r_w^2 \bar{t}_s / K_r$, $r_w = 0.05$ m, $K_r = 10^{-4}$ m/s and $S_s = 10^{-5}$ m$^{-1}$. The relation between
$\bar{t}_s$ and $\gamma$ can therefore be approximated as





$\log_{10} \bar{t}_s = \begin{cases} -\sum_{k=0}^{6} c_k (\log_{10} \gamma)^k & \text{for } 10^{-6} \leq \gamma \leq 2.7 \times 10^{-3} \\ 1 & \text{for } \gamma > 2.7 \times 10^{-3} \end{cases}$ (47)
where $c_0 = 629.90517$, $c_1 = 874.82145$, $c_2 = 500.07155$, $c_3 = 151.54284$, $c_4 = 25.63248$, $c_5 =$
$2.29276$, and $c_6 = 0.08471$ obtained by the Mathematica function Fit based on least-square
curve fitting. Existing models assuming hydraulic head response as SHM are applicable when
$\bar{t} \geq \bar{t}_s$ provided in Fig. 3 for a known value of $\gamma$.
**3.2. Radius of influence from pumping well**
Researchers have paid attention to the identification of aquifer hydraulic parameters within the
dimensionless radius of influence $\bar{R}$ from an oscillatory pumping well (e.g., Cadiff and Sayler.,
2016). This section quantifies $\bar{R}$ that is dominated by the magnitude of $\gamma$. Define $\bar{R}$ from the
pumping well to a location where $\bar{R}$ satisfies
$\bar{A}_t(\bar{R}, \bar{z}) = 10^{-2}$ (48)
where $\bar{A}_t$ is defined in Eq. (24d), $\bar{z}$ can be an arbitrary value of $0 \leq \bar{z} \leq 1$ because
$\bar{A}_t(\bar{R}, \bar{z})$ is independent of $\bar{z}$, and the value $10^{-2}$ causes an insignificant dimensional amplitude
that is defined as $Q\bar{A}_s(\bar{R}, \bar{z})/(2\pi b K_r)$ less than 1 mm for the practical ranges of $bK_r \geq 10^3$
m²/d and $Q \leq 10^2$ m³/d (Rasmussen et al. 2003). The Mathematica function FindRoot is
applied to solve Eq. (48) to determine the value of $\bar{R}$. Figure 4 shows the attenuation of the
amplitude $\bar{A}_t(\bar{r}, \bar{z})$ at $\bar{z} = 0.5$ for various values of $\gamma$ in panel (a) and the curve of $\gamma$ versus
$\bar{R}$ calculated by Eq. (48) in panel (b). The greater value of $\gamma$ causes smaller $\bar{A}_t$ and $\bar{R}$,
indicating that higher frequency of oscillatory pumping, larger aquifer storage or lower aquifer
horizontal conductivity leads to smaller amplitude of groundwater fluctuation and smaller
radius of influence. When $\gamma > 2.8 \times 10^{-2}$, the largest dimensionless amplitude at the rim of
the pumping well is less than $10^{-2}$ (i.e., $\bar{A}_t(1, \bar{z}) < 10^{-2}$). The magnitude of $\bar{R}$ can
therefore be considered as unity. The changes in $\kappa$ and $\sigma$ cause insignificant effect on the
estimates of $\bar{A}_t$ and $\bar{R}$. The magnitude of $\alpha$ related to wellbore storage effect will be discussed
in the next section. With the Mathematica function Fit, the relation between $\bar{R}$ and $\gamma$ can be
approximated as





$$\log_{10} \bar{R} = \begin{cases} \sum_{k=0}^{6} c_k (\log_{10} \gamma)^k & \text{for } 10^{-6} \le \gamma \le 2.8 \times 10^{-2} \\ 0 & \text{for } \gamma > 2.8 \times 10^{-2} \end{cases} \tag{49}$$
where $c_0 = -4.13203$, $c_1 = -2.83369$, $c_2 = 0.56905$, $c_3 = 0.65943$, $c_4 = 0.18209$, $c_5 = 0.02147$
and $c_6 = 9.33152 \times 10^{-4}$. It serves as a handy tool of estimating $\bar{R}$ within which observation
wells can receive signal from an oscillatory pumping well.
**3.3. Effect of wellbore storage on head fluctuation**
The effect of wellbore storage is dominated by the magnitude of $\alpha$ accounting for variation in
the well radius. This section discusses the discrepancy due to assuming an infinitesimal radius.
Figure 5 demonstrates the hydraulic head predicted by the present pseudo-steady state solution,
Eq. (26), for $\alpha = 10^{-2}$, $10^{-1}$, $1$, $10$, $10^2$ and $10^3$ at (a) $\bar{r} = 1$ at the rim of the pumping well and
(b) $\bar{r} = 16$ away from the well. The Dagan and Rabinovich (2014) solution assuming an
infinitesimal radius is taken for comparison. For the case of $\bar{r} = 1$, Fig. 5(a) indicates that the
predicted dimensionless amplitude increases with decreasing $\alpha$ and remains constant when $\alpha$
$\le 10^{-1}$. The Dagan and Rabinovich (2014) solution gives an overestimate of dimensionless
amplitude because of neglecting the wellbore storage effect. This result differs from the finding
of Papadopulos and Cooper (1967) that the effect is ignorable for a large time of a constant-
rate pumping test (i.e., $t > 2.5 \times 10^2 r_c^2 / (K_r b)$). For the case of $\bar{r} = 16$ (or $\bar{r} \ge 16$), both
solutions agree well when $\alpha \le 10$, indicating that the wellbore storage effect gradually
diminishes with distance from the pumping well. The effect should therefore be considered in
OPT models especially when observed hydrulic head data are taken close to the pumping well.
**3.4. Sensitivity analysis**
The normalized sensitivity coefficient $S_i$ defined as Eq. (44) with $X = h_{\exp}(r, z, t)$ in Eq.
(24b) is displayed in Fig. 6 for the response of exponential decay to the change in each of
parameters $K_r$, $K_z$, $S_s$, $S_y$ and $\omega$ with $\omega = $ (a) $2\pi/60$ s$^{-1}$ and (b) $2\pi/30$ s$^{-1}$. The figure indicates that
exponential decay is very sensitive to variation in each of $K_r$, $K_z$, $S_s$ and $\omega$ because of $|S_i| > 0$.
Precisely, a positive perturbation in $K_r$, $S_s$, and $\omega$ produces an increase in the magnitude of
$h_{\exp}(r, z, t)$ while that in $K_z$ causes a decrease. It is worth noting that the coefficient $S_i$ for $S_y$



is very close to zero over the entire period of time, indicating that $h_{\exp}(r,z,t)$ is insensitive
to the change in $S_y$ and the subtle change of gravity drainage has no influence on the exponential
decay. In addition, the sensitivity curves of $K_z$ and $S_s$ are symmetrical to the horizontal axis,
implying that these two parameters are highly correlated (Yeh and Chen, 2007). On the other
hand, the spatial distributions of the normalized sensitivity coefficient $S_i$ defined in Eq. (44)
with $X = A_t(r,z)$ in Eq. (24d) are shown in Fig. 7 for SHM amplitude in response to the
changes in parameters $K_r$, $K_z$, $S_s$, $S_y$ and $\omega$ for $\omega$ = (a) $2\pi/60$ s$^{-1}$ and (b) $2\pi/30$ s$^{-1}$. The figure
also indicates that $A_t(r,z)$ is sensitive to the change in each of $K_r$, $K_z$, $S_s$ and $\omega$ but insensitive
to the change in $S_y$. From those discussed above, we can conclude that the changes in the four
key parameters $K_r$, $K_z$, $S_s$ and $\omega$ significantly affect OPT model prediction, but the change in $S_y$
doesn't.
**3.5. Application of the present solution to field experiment**
Rabinovich et al. (2015) conducted a field OPT in an unconfined aquifer at the BHRS. The
aquifer contains a mix of sand, gravel and cobble sediments with 20 m averaged thickness. The
aquifer bottom is a clay confining unit. The pumping well fully penetrating the aquifer has 10
cm inner diameter and 11.43 cm outer diameter of PVC casing. The pumping rate can be
approximated as $Q\sin(\omega t)$ with $Q = 5.8\times10^{-5}$ m$^3$/s and $\omega = 2\pi/24$ s$^{-1}$. The observation data
of SHM representing time-varying hydraulic head at the pumping well after a certain period of
time are plotted in Fig. 8.
The aquifer hydraulic parameters $K_r$, $K_z$, $S_s$, and $S_y$ can be determined by the pseudo-steady
state solutions, Eqs. (38a) and (43a), coupled with the Levenberg–Marquardt algorithm
provided in the Mathematica function FindFit (Wolfram, 1991). Define the residual sum of
square (RSS) as $\text{RSS} = \sum_{i=1}^{m} e_i^2$ and the mean error (ME) as $\text{ME} = \frac{1}{m}\sum_{i=1}^{m} e_i$ where $e_i$ is the
difference between predicted and observed hydraulic heads and $m$ is the number of observation
data (Yeh, 1987). The estimated parameters are $K_r = 1.034\times10^{-5}$ m/s, $K_z = 1.016\times10^{-5}$ m/s, $S_s$
$= 8.706\times10^{-5}$ m$^{-1}$, $S_y = 5.708\times10^{-3}$ with RSS $= 1.184\times10^{-3}$ m$^2$ and ME $= 0.5718$ m for the case





of unconfined aquifers and $K_r = 5.035 \times 10^{-4}$ m/s, $S_s = 1.40998 \times 10^{-5}$ 1/m with RSS = $7.454 \times 10^{-}$
$^4$ m$^2$ and ME = 0.46683 m for the case of confined aquifers. The estimated $S_y$ is less than two
orders of the typical range of 0.01~0.3 (Freeze and Cherry, 1979), which accords with the
findings of Rasmussen et al. (2003) and Rabinovich et al. (2015). One reason for an
underestimated $S_y$ may be because flow behaviors associated with OPT and constant-rate
pumping test are different especially for a high frequency (i.e., $\omega$). The moisture exchange was
limited by capillary fringe between the zones below and upper the water table. Several
laboratory researches have focused on this subject for a short period or high frequency of an
oscillatory pumping test (e.g., Cartwright et al., 2003; 2005) and they confirmed that the values
of $S_y$ decreases more than two orders at small period of oscillation, compared with conventional
instantaneous drainage.

Rabinovich et al. (2015) reported $K_r = 6.3833 \times 10^{-4}$ m/s, $S_s = 9.22 \times 10^{-6}$ 1/m, $S_y =$

$8.691 \times 10^{-4}$ with RSS = $2.638 \times 10^{-3}$ m$^2$ and ME = 0.5955 m for the case of unconfined aquifers
and $K_r = 7.149 \times 10^{-4}$ m/s, $S_s = 1.214 \times 10^{-5}$ 1/m with RSS = $3.992 \times 10^{-3}$ m$^2$ and ME = 0.5958 m
for the case of confined aquifers on the basis of the Dagan and Rabinovich (2014) solution.
Our work provides smaller RSSs than theirs. This may be attributed to the fact that the present
solution considers the effect of wellbore storage on the parameter determination. Figure 8
displays agreement between the observation data and the head fluctuations predicted by the
pseudo-steady state solution, Eq. (38a), for unconfined aquifers and Eq. (43a) for confined
aquifers based on those estimated parameters. This indicates that the present solution is
applicable to real-world OPT.
**4. Concluding remarks**
A variety of analytical solutions have been proposed so far, but little attention is paid to the
combined effects of wellbore storage and initial condition before OPT. This study develops a
new model for describing hydraulic head fluctuation due to OPT in unconfined aquifers. Static
hydraulic head prior to OPT is regarded as an initial condition. An equation accounting for





wellbore storage effect is specified at the rim of a finite-radius pumping well. A linearized free
surface equation is considered as the top boundary condition. The analytical solution of the
model is derived by the Laplace transform and finite integral transform. The sensitivity analysis
of the head response to the change in each of hydraulic parameters is performed. The present
solution can estimate aquifer hydraulic parameters when coupling the Levenberg–Marquardt
algorithm and observation data. Our findings are summarized below:
1.    The transient solution of dimensionless hydraulic head is expressed as the sum of the

exponential and harmonic functions of time (i.e., $\bar{h} = \bar{h}_{\exp} + \bar{h}_{\mathrm{SHM}}$) in Eq. (24a) or (25a).

The latter function can be replaced by the pseudo-steady state solution with error less than

3%.

2.    The exponential function $\bar{h}_{\exp}$ defined in Eq. (24b) or (25b) accounts for the effect of the

initial condition of static groundwater prior to OPT. The effect diminishes when $\bar{t} \geq \bar{t}_s$

that can be approximated by Eq. (47) for a fixed dimensionless frequency $\gamma$. Existing

analytical solutions assuming SHM without the initial condition are applicable when the

condition $\bar{t} \geq \bar{t}_s$ is met.

3.    The magnitudes of $\alpha$ and $\bar{r}$ dominate the influence of wellbore storage on predicted head

fluctuation due to OPT. Neglecting the influence causes a significant overestimate of the

amplitude of SHM at the pumping well (i.e., $\bar{r} = 1$) in spite of an extreme range $\alpha \leq 10^{-1}$

for very small well radius. In contrast, the influence gradually diminishes with distance

from the pumping well and is ignorable when $\bar{r} \geq 16$ and $\alpha \leq 10$. Existing analytical

solutions assuming an infinitesimal radius can predict accurate head fluctuation when these

two conditions are met.

4.    The dimensionless radius of influence $\bar{R}$ can be estimated by Eq. (49) with a

dimensionless frequency $\gamma$. Observation wells should be located in the area of $\bar{r} < \bar{R}$ for

obtaining observable data of head fluctuations.

5.    The sensitivity analysis suggests that the changes in four parameters $K_r$, $K_z$, $S_s$ and $\omega$





significantly affect OPT model prediction but that in $S_y$ doesn't exert any effect.

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

**Acknowledgments**
Research leading to this paper has been partially supported by the grants from the Fundamental
Research Funds for the Central Universities (2018B00114), the National Natural Science
Foundation of China (41561134016 and 51421006) and the Taiwan Ministry of Science and
Technology under the contract numbers MOST 105-2221-E-009 -043-MY2 and MOST 106-
2221-E-009 -066. The authors would like to thank Prof. John L. Wilson for his suggestion of
including the wellbore effect in the OPT model.

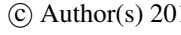



**Appendix A: Finite integral transform**
Applying the finite integral transform to the model of Eqs. (14) – (18) results in (Latinopoulos,

1985)

$\tilde{h}(\beta_n) = \Im\{\hat{h}(\bar{z})\} = \int_0^1 \hat{h}(\bar{z}) \, F_t \cos(\beta_n \bar{z}) \, d\bar{z}$ (A.1)
$F_t = \left(\frac{2(\beta_n^2 + a^2 p^2)}{\beta_n^2 + a^2 p^2 + ap}\right)^{0.5}$ (A.2)
where $\beta_n$ is the root of Eq. (19). On the basis of integration by parts, one can write
$\Im\left\{\frac{\partial^2 \hat{h}}{\partial \bar{z}^2}\right\} = \int_0^1 \left(\frac{\partial^2 \hat{h}}{\partial \bar{z}^2}\right) F(\beta_n) \cos(\beta_n z) \, d\bar{z} = -\beta_n^2 \tilde{h}$ (A.3)
Note that Eq. (A.3) is applicable only for the no-flow condition specified at $\bar{z} = 0$ (i.e., Eq.
(17)) and third-type condition specified at $\bar{z} = 1$ (i.e., Eq. (16)). The formula for the inverse
finite integral transform is defined as
$\hat{h}(\bar{z}) = \Im^{-1}\{\tilde{h}(\beta_n)\} = \sum_{n=1}^{\infty} \tilde{h}(\beta_n) F(\beta_n) \cos(\beta_n \bar{z})$ (A.4)
Similarly, apply the transform to the model of Eqs. (27) – (31); one can have
$\tilde{H}(\beta_m) = \Im\{\bar{H}(\bar{z})\} = \int_0^1 \bar{H}(\bar{z}) \, F_s \cos(\beta_m \bar{z}) \, d\bar{z}$ (A.5)
where $F_s$ is defined in Eq. (35); $\beta_m$ is the root of Eq. (36). It also has the property that
$\Im\left\{\frac{\partial^2 \bar{H}}{\partial \bar{z}^2}\right\} = \int_0^1 \left(\frac{\partial^2 \bar{H}}{\partial \bar{z}^2}\right) F_s \cos(\beta_m z) \, d\bar{z} = -\beta_m^2 \tilde{H}$ (A.6)
Again, Eq. (A.6) is applicable only for the no-flow condition specified at $\bar{z} = 0$ (i.e., Eq. (30))
and third-type condition specified at $\bar{z} = 1$ (i.e., Eq. (29)). The inverse finite integral
transform can be written as
$\bar{H}(\bar{z}) = \Im^{-1}\{\tilde{H}(\beta_m)\} = \sum_{m=1}^{\infty} \tilde{H}(\beta_m) F_s \cos(\beta_m \bar{z})$ (A.7)





## Appendix B: Derivation of Eqs. (24a) – (24k)

On the basis of Eq. (A.4) and taking the inverse finite integral transform to Eq. (22), one can

have the Laplace-domain solution as

$$\hat{h}(\bar{r}, \bar{z}, p) = 2 \sum_{n=1}^{\infty} \tilde{h}(\bar{r}, \beta_n, p) \cos(\beta_n \bar{z}) \qquad \text{(B.1)}$$

with

$$\tilde{h}(\bar{r}, \beta_n, p) = \hat{h}_1(p) \cdot \hat{h}_2(p) \qquad \text{(B.2)}$$

$$\hat{h}_1(p) = \frac{\gamma}{(p^2 + \gamma^2)} \qquad \text{(B.3)}$$

$$\hat{h}_2(p) = -\varphi_1 \varphi_2 \qquad \text{(B.4)}$$

$$\varphi_1 = \sin \beta_n \, K_0(\bar{r}\lambda) / \left( \beta_n \big( p\alpha K_0(\lambda) + \lambda K_1(\lambda) \big) \right) \qquad \text{(B.5)}$$

$$\varphi_2 = (\beta_n^2 + a^2 p^2) / (\beta_n^2 + a^2 p^2 + ap) \qquad \text{(B.6)}$$

where $\lambda$ is defined in Eq. (23). Using the Mathematica function InverseLaplaceTransform, the

inverse Laplace transform for $\hat{h}_{p1}(p)$ in Eq. (B.3) can be obtained as

$$\hat{h}_1(\bar{t}) = \sin(\gamma \, \bar{t}) \qquad \text{(B.7)}$$

The inverse Laplace transform for $\hat{h}_{p2}(\bar{r}, \beta_n, p)$ in Eq. (B.4) is defined as

$$\hat{h}_2(\bar{t}) = \frac{1}{2\pi i} \int_{\xi - i\infty}^{\xi + i\infty} \hat{h}_2(p) \, e^{p\bar{t}} dp \qquad \text{(B.8)}$$

where $\xi$ is a real number being large enough so that all singularities are on the LHS of the

straight line from $(\xi, -i\infty)$ to $(\xi, i\infty)$ in the complex plane. The integrand $\hat{h}_2(p)$ is a

multiple-value function with a branch point at $p = -\mu \beta_n^2$ and a branch cut from the point

along the negative real axis. In order to reduce $\hat{h}_2(p)$ to a single-value function, we consider

a modified Bromwich contour that contains a straight line $\overline{AB}$, $\overline{CD}$ right above the branch cut

and $\overline{EF}$ right below the branch cut, a semicircle with radius $R$, and a circle $\overset{\frown}{DE}$ with radius $\varepsilon$

in Fig. A1. According to the residual theory and the Bromwich integral, Eq. (B.8) becomes

$$\hat{h}_2(\bar{t}) + \lim_{\substack{\varepsilon \to 0 \\ R \to \infty}} \frac{1}{2\pi i} \left[ \int_B^C \hat{h}_2(p) \, e^{p\bar{t}} dp + \int_C^D \hat{h}_2(p) \, e^{p\bar{t}} dp + \int_D^E \hat{h}_2(p) \, e^{p\bar{t}} dp + \right.$$

$$\left. \int_E^F \hat{h}_2(p) \, e^{p\bar{t}} dp + \int_F^A \hat{h}_2(p) \, e^{p\bar{t}} dp \right] = 0 \qquad \text{(B.10)}$$

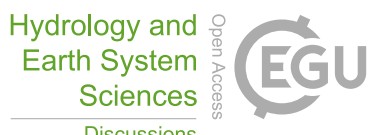

where zero on the RHS is due to no pole in the complex plane. The integrations for paths $\overset{\frown}{BA}$
(i.e. $\int_B^C \hat{h}_2(p)\, e^{p\bar{t}}dp + \int_F^A \hat{h}_2(p)\, e^{p\bar{t}}dp$) with $R \to \infty$ and $\overset{\frown}{DE}$ (i.e. $\int_D^E \hat{h}_2(p)\, e^{p\bar{t}}dp$) with
$\varepsilon \to 0$ equal zero. The path $\overline{CD}$ starts from $p = -\infty$ to $p = -\mu\beta_n^2$ and $\overline{EF}$ starts from
$p = -\mu\beta_n^2$ to $p = -\infty$. Eq. (B.10) therefore reduces to
$\hat{h}_2(\bar{t}) = -\frac{1}{2\pi i}\left(\int_{-\infty}^{-\mu\beta_n^2} \hat{h}_2(p^+)e^{p^+\bar{t}}dp + \int_{-\mu\beta_n^2}^{-\infty} \hat{h}_2(p^-)e^{p^-\bar{t}}dp\right)$     (B.11)
where $p^+$ and $p^-$ are complex numbers right above and below the real axis, respectively.
Consider $p^+ = \zeta e^{i\pi} - \mu\beta_n^2$ and $p^- = \zeta e^{-i\pi} - \mu\beta_n^2$ in the polar coordinate system with the
origin at $(-\mu\beta_n^2, 0)$. Eq. (B.11) then becomes
$\hat{h}_2(\bar{t}) = \frac{-1}{2\pi i}\int_0^\infty \hat{h}_2(p^+)e^{p^+\bar{t}}dp - \hat{h}_2(p^-)e^{p^-\bar{t}}d\zeta$     (B12)
where $p^+$ and $p^-$ lead to the same result of $p_0 = -\zeta - \mu\beta_n^2$ for a given $\zeta$; $\lambda = \sqrt{p + \mu\beta_n^2}$
equals $\lambda_0 = \sqrt{\zeta}i$ for $p = p^+$ and $-\lambda_0$ for $p = p^-$. Note that $\hat{h}_2(p^+)\,e^{p^+\bar{t}}$ and
$\hat{h}_2(p^-)\,e^{p^-\bar{t}}$ are in terms of complex numbers. The numerical result of the integrand in Eq.
(B.12) must be a pure imaginary number that is exactly twice of the imaginary part of a complex
number from $\hat{h}_2(p^+)\,e^{p^+t}$ with $p^+ = p_0$ and $\lambda = \lambda_0$. The inverse Laplace transform for
$\hat{h}_2(p)$ can be written as
$\hat{h}_2(\bar{t}) = \frac{-1}{\pi}\int_0^\infty \mathrm{Im}\big(\varphi_1\varepsilon_2\, e^{p_0\bar{t}}\big)\, d\zeta$     (B.13)
where $p = p_0$; $\lambda = \lambda_0$; $\varphi_1$ and $\varepsilon_2$ are respectively defined in Eqs. (B.5) and (24i); Im(·)
represents the numerical imaginary part of the integrand. According to the convolution theory,
the inverse Laplace transform for $\tilde{h}(\bar{r}, \beta_n, p)$ is
$\hat{h}(\bar{r}, \beta_n, \bar{t}) = \int_0^t \hat{h}_2(\tau)\, \hat{h}_1(\bar{t} - \tau)d\tau$     (B.14)
where $\bar{h}_1(\bar{t} - \tau) = \sin(\gamma(\bar{t} - \tau))$ based on Eq. (B.7); $\bar{h}_2(\tau)$ is defined in Eq. (B.13) with
$\bar{t} = \tau$. Eq. (B.14) can reduce to
$\hat{h}(\bar{r}, \beta_n, \bar{t}) = \frac{-1}{\pi}\int_0^\infty \mathrm{Im}\left(\frac{\varphi_1\varepsilon_{22}(\gamma e^{p_0\bar{t}} - \gamma\cos(\gamma\bar{t}) - p_0\sin(\gamma\bar{t}))}{p_0^2 + \gamma^2}\right)d\zeta$     (B.15)





Substituting $\tilde{h}(\bar{r}, \beta_n, p) = \hat{h}(\bar{r}, \beta_n, \bar{t})$ and $\hat{h}(\bar{r}, \bar{z}, p) = \bar{h}(\bar{r}, \bar{z}, \bar{t})$ into Eq. (B.1) and
rearranging the result leads to
$\bar{h}(\bar{r}, \bar{z}, \bar{t}) = \frac{-2}{\pi} \sum_{n=1}^{\infty} \int_0^{\infty} \cos(\beta_n \bar{z}) \, \mathrm{Im}\left(\varepsilon_1 \varepsilon_2 \gamma e^{p_0 \bar{t}}\right) d\zeta +$

$\frac{2}{\pi} \sum_{n=1}^{\infty} \int_0^{\infty} \cos(\beta_n \bar{z}) \, \mathrm{Im}\left(\varepsilon_1 \varepsilon_2 (\gamma \cos(\gamma \bar{t}) + p_0 \sin(\gamma \bar{t}))\right) d\zeta$       (B.16)

where $\varepsilon_1$ and $\varepsilon_2$ are defined in Eqs. (24h) and (24i); the first RHS term equals $\bar{h}_{\mathrm{exp}}(\bar{r}, \bar{z}, \bar{t})$
defined in Eq. (24b); the second term can be expressed as $\bar{h}_{\mathrm{SHM}}(\bar{r}, \bar{z}, \bar{t})$ defined in Eq. (24c).
Finally, the complete solution is expressed as Eqs. (24a) – (24k).



**Figures**




**Figure 1.** Schematic diagram for an oscillatory pumping test at a fully penetrating well of
finite radius in an unconfined aquifer





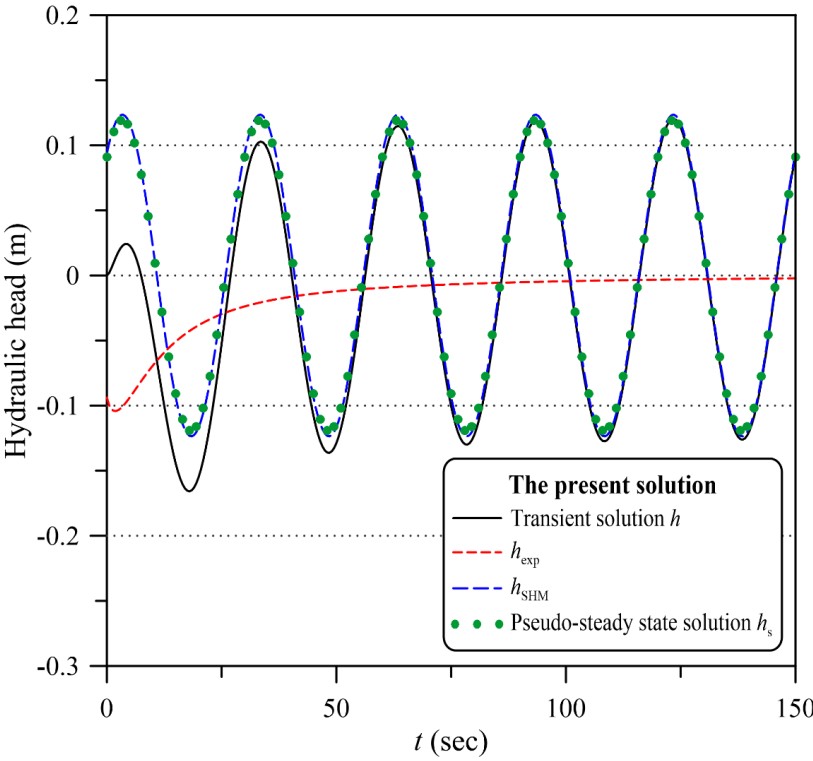


**Figure 2.** Hydraulic head predicted by the transient solution expressed as $h = h_{\mathrm{exp}} + h_{\mathrm{SHM}}$
and the pseudo-steady state solution $h_{\mathrm{s}}$ for unconfined aquifers





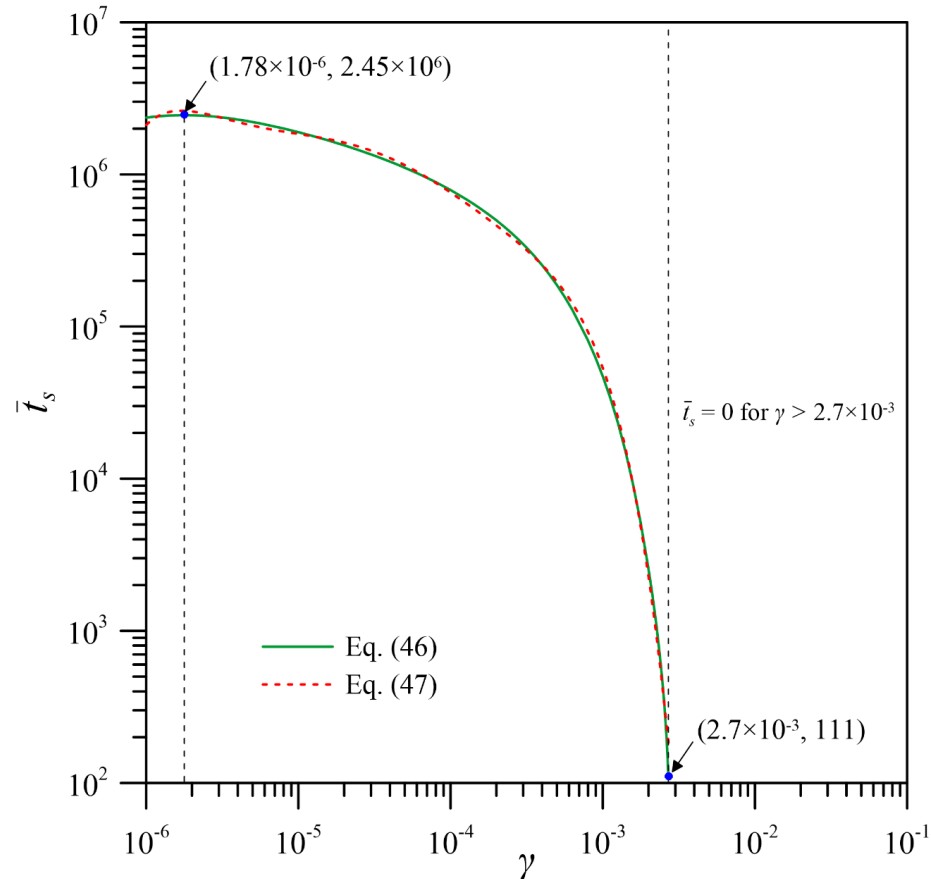


**Figure 3.** The curve of dimensionless frequency $\gamma$ of oscillatory pumping rate versus the

dimensionless time at which hydraulic head fluctuation can be regarded as SHM



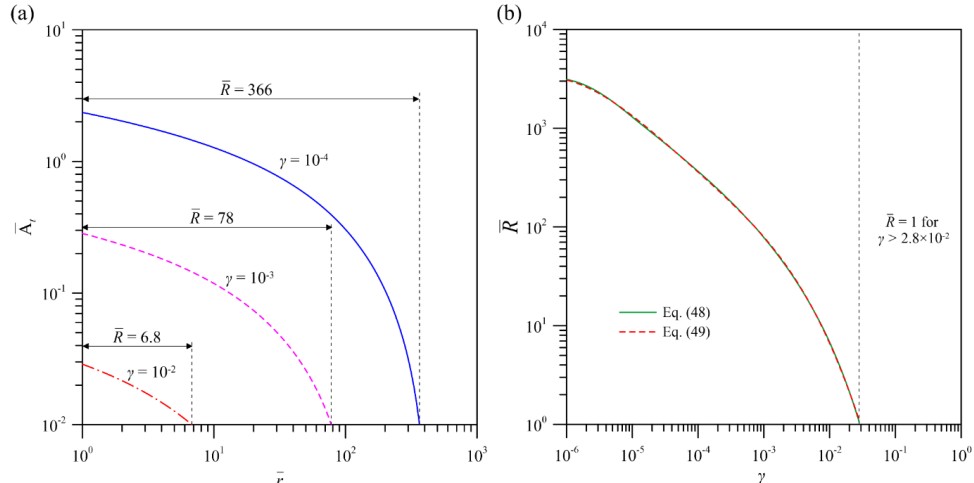

**Figure 4.** (a) Attenuation of dimensionless amplitude and (b) dimensionless radius of influence for different dimensionless frequency $\gamma$ of oscillatory pumping rate for unconfined aquifers





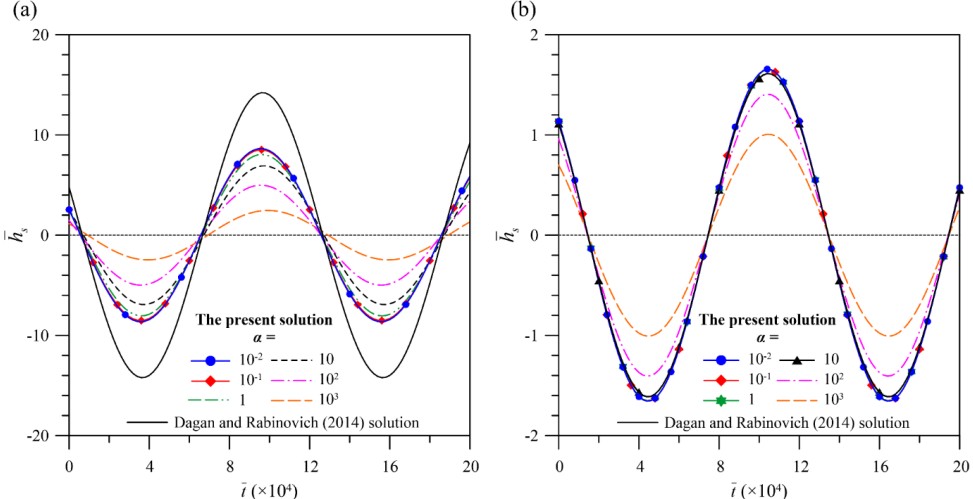

**Figure 5.** Predicted Head fluctuations for (a) $\bar{r} = 1$ at the rim of the pumping well and (b) $\bar{r}$

= 16 away from the well using the Dagan and Rabinovich (2014) solution and the present

solution with different $\alpha$ related to wellbore storage effect for unconfined aquifers





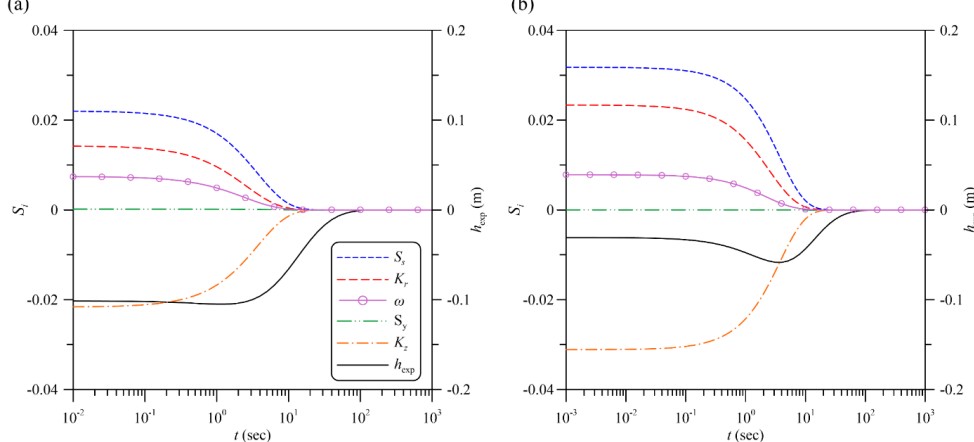

**Figure 6.** Temporal distributions of the normalized sensitivity coefficient $S_i$ associated with the exponential component defined in Eq. (24b) for parameters $K_r$, $K_z$, $S_s$, $S_y$ and $\omega$ when $\omega =$ (a) $2\pi/60$ s$^{-1}$ and (b) $2\pi/30$ s$^{-1}$





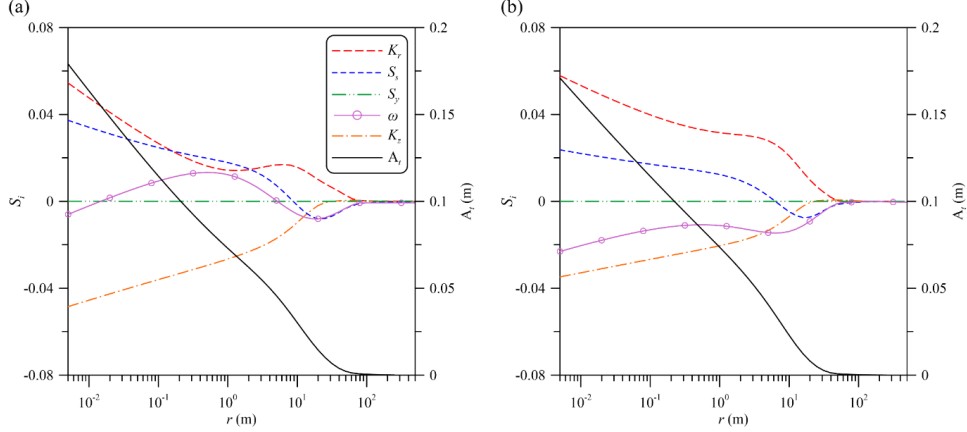


**Figure 7.** Spatial distributions of the normalized sensitivity coefficient $S_i$ associated with


SHM amplitude defined in Eq. (24d) for each of parameters $K_r$, $K_z$, $S_s$, $S_y$, and $\omega$ when $\omega$ = (a)


$2\pi/60$ s$^{-1}$ and (b) $2\pi/30$ s$^{-1}$




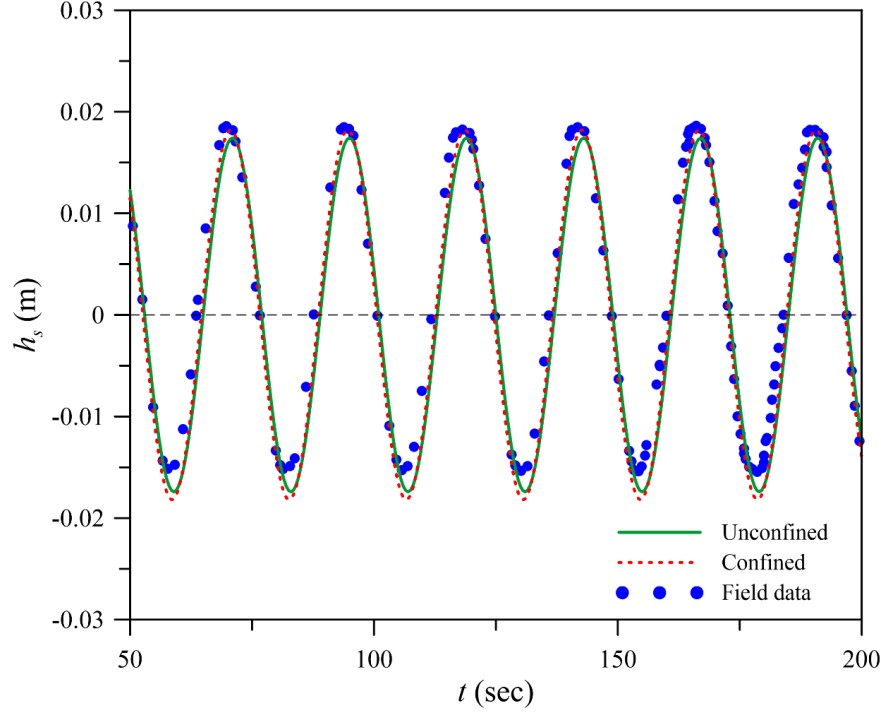


**Figure 8.** Comparision of field observation data with head fluctuations predicted by the
pseudo-steady state solutions Eq. (38a) for unconfined aquifers and Eq. (43a) for confined
aquifers





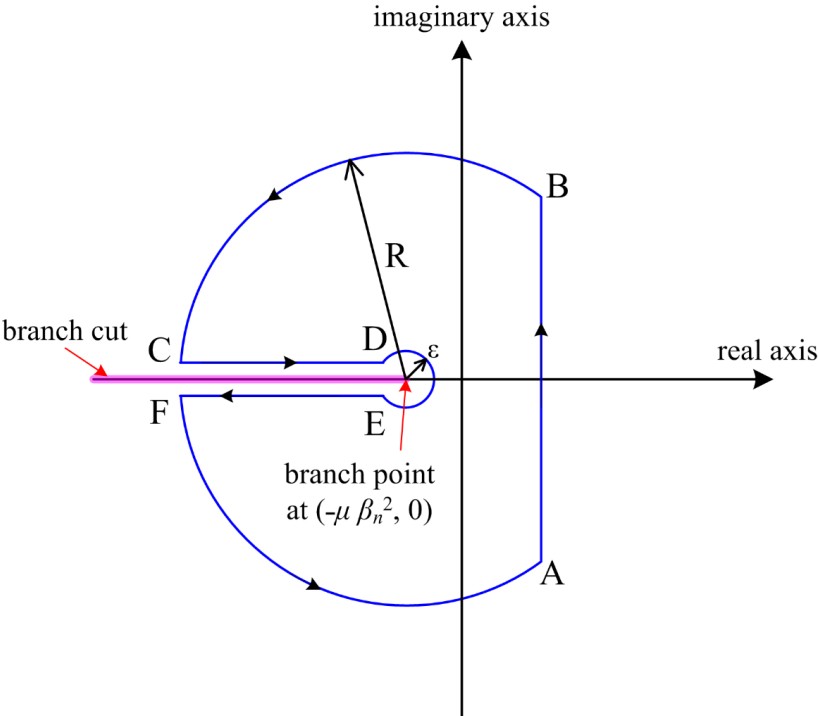


**Figure A1.** Modified Bromwich contour for the inverse Laplace transform to a multiple-value
function with a branch point and a branch cut