# Peer review of "Analysis of Groundwater Response to Oscillatory Pumping Test in Unconfined Aquifers: Consider the Effects of Initial Condition and Wellbore Storage"

_Hydrology and Earth System Sciences, 2018_

## Referee Comment (RC1) · K. Halford (Referee) · 30 May 2018

**United States Department of the Interior**

**U. S. GEOLOGICAL SURVEY**

2730 N. Deer Run Rd.
Carson City, NV
Phone: 775-887-7614

May 25, 2018

Graham Fogg, Editor, *Hydrology and Earth System Sciences*

Dear Graham:

I have reviewed manuscript hess-2018-199, "Analysis of Groundwater Response to Oscillatory Pumping Test in Unconfined Aquifers: Consider the Effects of Initial Condition and Wellbore Storage" by Ching-Sheng Huanga, Ya-Hsin Tsai, Hund-Der Yeh and Tao Yang. The paper offers another analytical solution for simulating oscillatory pumping, which allegedly will make this a viable aquifer-testing approach. Oscillatory pumping is limited more by complicated field equipment when compared to slug tests. Results from oscillatory pumping would need to be significantly better than slug-test results for me to suffer the additional logistical burden. This paper, like several others, does not demonstrate an advantage of oscillatory pumping to functionally similar slug tests, so slug tests are compared to oscillatory pumping in this review and found superior. Hydrologists really do not need this paper or any others about oscillatory pumping.

Oscillatory-pumping, aquifer tests have practical limitations such as small volumes of displaced water and complex methods of displacement. Small volumes clearly limit the volume of aquifer investigated, which practically limits the method to single-well tests. Small pumping volumes also significantly increase the uncertainty of flow across the well screen. This is because wellbore storage in the pumping well is similar to the volume of water pumped. Oscillatory pumping, like slug tests, remain sensitive to unknown entry losses from wellbore damage.

Oscillatory pumping and slug tests are investigated and compared with a two-dimensional, radial flow model of a shallow, unconfined aquifer that was simulated with MODFLOW (Harbaugh, 2005). The model extended vertically from 0 to 30 ft above the base of the aquifer, where the upper row of the model was the water table, and the 30-ft thickness was divided into uniform, 1-ft thick layers. Temporal changes in the saturated thickness of the aquifer were not simulated because the maximum drawdown near the water table was small relative to total thickness. The model grid was divided into 30 rows of 49 columns that were centered on the pumping well and extended away 37,500 ft. Horizontal hydraulic conductivities ($K_x$) of 0.1, 3, and 100 ft/d for the aquifer were simulated. Horizontal-to-vertical anisotropy, specific storage, and specific yield of 10 d'less, 2.E-6 1/ft, and 0.1 d'less, respectively, were assigned.

Oscillatory pumping from a 3-in diameter well in a 6-in borehole with wellbore storage and wellbore damage was simulated. Oscillatory pumping was approximated by 1-second stress periods during a 24-second cycle, where the peak flow rate was 0.92 gpm (5.8E-05 m³/s). Twelve cycles were simulated and water-level changes from the last two cycles were presented (Figure 1KJH).  The pumping well was simulated with the first column as 10 ft of cells with very high hydraulic conductivities, a specific yield of 1, and specific storages of 0. Gravel pack of the annular space between well screen and formation was simulated with the second column and extended from 5 to 10 ft below the water table. Hydraulic conductivity of the gravel pack (Kann) ranged from 0.1 ft/d for significant wellbore damage to 100 ft/d for a fully developed well.

[Figure]

**Figure 1KJH.—Water-level changes in an oscillatory pumping well for horizontal hydraulic conductivities (Kx) of 0.1, 3, and 100 ft/d and annular hydraulic conductivities (Kann) of 0.1, 3, and 100 ft/d.**

Horizontal hydraulic conductivities between 1 and 300 ft/d could reasonably be estimated with oscillatory pumping. Water-level changes for Kx = 3 ft/d were best suited to analysis with the initial displacement largely dissipated and wellbore damage minimally affecting estimates (Figure 1KJH). Water-level changes for Kx = 100 ft/d were sensitive to wellbore damage, which causes Kx to be underestimated. Measurements of water-level changes approach inherent noise in a pumped well as Kx exceeds 300 ft/d. Slug tests are limited equally by wellbore damage and an upper threshold for estimating Kx. Water-level changes for Kx = 0.1 ft/d show a greater sensitivity to dissipating the initial displacement than amplitude of water-level changes. Amplitude was 0.31 ft for both Kx of 0.1 and 3 ft/d (Figure 1KJH), which suggests the oscillatory signal is insensitive to Kx < 1 ft/d. Slug tests are more appropriate for small hydraulic conductivities because recovery time is the limiting factor. About 100 minutes is the 90-% recovery time for a slug test in the example with Kx = 0.1 ft/d. Reasonable variations in well construction and peak pumping rates will not alter these conclusions.

Slug tests investigate significantly larger volumes of aquifer than oscillatory pumping (Figure 2KJH). An initial displacement of 3 ft (1.1 gallons in a 3-in well) simulated a slug test where Kx = 3 ft/d. Water levels were displaced at least 0.02 ft through the thickness of aquifer and about 50 ft away from the slug test.  Maximum displacements greater than 0.01 ft from oscillatory pumping mapped where amplitudes exceeded 0.02 ft and were comparable to maximum extents for slug tests. Oscillatory pumping influenced about 2 percent of the aquifer that was influenced by a typical slug test (Figure 2KJH).

[Figure]

**Figure 2KJH.—Maximum extent of water-level displacement from a slug test with 3 ft (1.1 gal) of initial displacement and oscillatory pumping with a peak rate of 0.92 gpm, where Kx = 3ft/d and Kann = 100 ft/d.**

The need for oscillatory pumping as an aquifer-testing method seems limited, which has greatly dampened my enthusiasm for reviewing anymore articles about oscillatory pumping.

Sincerely,  Keith J Halford

---

## Referee Comment (RC2) · M. Cardiff (Referee) · 1 Jun 2018

Review of: "Analysis of Groundwater Response to Oscillatory Pumping Test in Unconfined Aquifers: Consider the Effects of Initial Condition and Wellbore Storage" By Ching-Sheng Huang, Ya-Hsin Tsai, Hun-Der Yeh, and Tao Yang

Review by: Michael Cardiff

This paper is not acceptable in its present format for at least a few reasons. My primary reason is this: the authors have claimed to have used data from oscillatory pumping

tests (data collected at the Boise Hydrogeophysical Research Site (BHRS), by myself and colleagues). Looking at the data they claim to fit, I can guarantee it is not raw data from any of the tests we collected. As far as I am aware, the authors of this paper did not contact any of the primary collectors of this data in an effort to understand it, nor did they apply an analysis strategy that is appropriate. Publishing data that is suspect under the name of the workers from the BHRS (and using a flawed analysis to do so) negatively impacts those who have worked so hard to collect the high quality data available from this site.

The current paper claims to develop a novel method for analyzing fully-penetrating oscillatory tests in which wellbore storage and the water table are taken into account. Applying this model to our data from the BHRS is completely nonsensical because: 1) While the wellbore we pumped from was indeed fully penetrating, the wellbore was packed off above and below our "oscillation zone", meaning that only a 1m interval (partially penetrating) zone served as the pumping interval. This does not fit with the model that has been developed in this paper; also 2) There is no need to consider wellbore storage for the tests performed at Boise because we used a piston to generate the signal within the well (i.e., the oscillating zone was under confined conditions, and we forced water into / out of the formation via piston). For both these reasons, the model the authors have developed is inappropriate for analyzing our data. The authors may have found this out earlier had they bothered to contact any of the field workers who spent such time and effort collecting this data.

With regards to the scientific merit / value of the model itself – I also question whether this model is necessary or useful, and whether it is being considered for reasonable ranges of the given parameters. Consider Figure 5 – Figure 5(b) shows somewhat of a difference from the Dagan and Rabinovich solution at a distance of $\bar{r} = 16$. Given the non-dimensionalization used, this means it is at a distance of 16 well radii. A standard well radius is about 5 cm, meaning that this effect is being observed only at a distance of less than 0.8m from the pumping location. I have never in my life seen wells spaced

80cm apart. A very big well might be 20cm, for which the effect would apparently decay after only 3.2m.

The authors seem to have chosen parameters that are unrealistic for most aquifers. For example, they use a specific yield value of $S_y = 0.1$. Specific yield values in aquifer pumping tests have almost never been measured to be this high (due to delayed drainage), and in the special case of oscillator tests where saturation changes rapidly, it is unlikely even partial drainage will occur. Similarly, many of the other choices in the plots are suspect. Looking at the definition of $\alpha = \frac{r_c^2}{2r_w^2 S_s b}$, for example, I find it hard to understand why the authors have focused on cases such as $\alpha = 1$ and below in Figure 5. Given that $S_s$ is generally in the range of $10^{-5}m^{-1}$ to $10^{-6}m^{-1}$ for any natural material, and that reasonable aquifers may be 10-1000m thick, can one imagine any realistic solutions where $\alpha < 1$?

It is also notable that in Figure 8, the confined solution appears to fit the data perfectly well (using the same K and $S_s$ parameters as the unconfined solution, if I am reading correctly) almost exactly as well as the more complex model. This would indicate to me that the details considered in this more complex model matter not one bit, and the water table can simply be considered as a no-flux boundary practically in these tests.

Similarly, Figure 5(a) represents head at the edge of the wellbore itself, which is unlikely to be used in real field scenarios because measurements at the pumping location are subject to numerous nuisance factors (for example, wellbore "skin", non-darcian flow conditions near the wellbore, inertial effects, etc. So I see no practical reason to consider the variability in this result.

While it is mathematically interesting to derive new PDE solutions, I fail to see the practical application of these much more complex solutions, given that they are still invoking many assumptions / approximations. For example, the authors do not deal with the fact that they are using only an approximation for the water table response (the linearized free surface condition of Neumann), and that realistically oscillatory tests are

likely to be subject to delayed drainage and differing yields as a function of frequency. For all of these reasons I cannot recommend that this paper be published.

---

## Author Comment (AC1) · 10 Jun 2018

**Reply to Reviewer 1**

Dear Graham:

I have reviewed manuscript hess-2018-199, "Analysis of Groundwater Response to Oscillatory Pumping Test in Unconfined Aquifers: Consider the Effects of Initial Condition and Wellbore Storage" by Ching-Sheng Huanga, Ya-Hsin Tsai, Hund-Der Yeh and Tao Yang. The paper offers another analytical solution for simulating oscillatory pumping, which allegedly will make this a viable aquifer-testing approach. Oscillatory pumping is limited more by complicated field equipment when compared to slug tests. Results from oscillatory pumping would need to be significantly better than slug-test results for me to suffer the additional logistical burden. This paper, like several others, does not demonstrate an advantage of oscillatory pumping to functionally similar slug tests, so slug tests are compared to oscillatory pumping in this review and found superior. Hydrologists really do not need this paper or any others about oscillatory pumping.

Response: Over the past few decades, a considerable number of studies have been made on the oscillatory pumping test (OPT) (e.g., Black and Kipp, 1981; Cardiff et al., 2013; Dagan and Rabinovich, 2014; Rabinovich et al., 2015). The OPT has at least following two advantages (Bakhos et al., 2014) over the slug test:

(1) The hydraulic head is sensitive to external changes, such as changes in the level of rivers adjacent to the field area, pumping or irrigation near observation wells, tidal effect, barometric pressure, changes in overburden. Noise from these sources may affect results in a variety of ways (Spane and Mackley, 2011). Therefore, an important merit of OPT is the ability to extract the signal from a variety of different types of noise, even when the signal is much smaller than the level of noise, provided the duration of OPT is long enough.

(2) Pseudo-steady state models for OPT reduce computational cost as mentioned in the comment.

Oscillatory-pumping, aquifer tests have practical limitations such as small volumes of displaced water and complex methods of displacement.   Small volumes clearly limit the volume of aquifer

investigated, which practically limits the method to single-well tests. Small pumping volumes also significantly increase the uncertainty of flow across the well screen. This is because wellbore storage in the pumping well is similar to the volume of water pumped. Oscillatory pumping, like slug tests, remain sensitive to unknown entry losses from wellbore damage.

Response: We agree to the comment on the limitations of OPT. However, the OPT has its abovementioned advantages that traditional pumping tests cannot achieve. Oscillatory piston pumping test (OPPT) can significantly diminish wellbore storage effect as addressed by reviewer #2 so that the radius of influence from the pumping well can largely extend.

Oscillatory pumping and slug tests are investigated and compared with a two-dimensional, radial flow model of a shallow, unconfined aquifer that was simulated with MODFLOW (Harbaugh, 2005). The model extended vertically from 0 to 30 ft above the base of the aquifer, where the upper row of the model was the water table, and the 30-ft thickness was divided into uniform, 1-ft thick layers. Temporal changes in the saturated thickness of the aquifer were not simulated because the maximum drawdown near the water table was small relative to total thickness. The model grid was divided into 30 rows of 49 columns that were centered on the pumping well and extended away 37,500 ft. Horizontal hydraulic conductivities (Kx) of 0.1, 3, and 100 ft/d for the aquifer were simulated. Horizontal-to-vertical anisotropy, specific storage, and specific yield of 10 d'less, 2.E-6 1/ft, and 0.1 d'less, respectively, were assigned.

Oscillatory pumping from a 3-in diameter well in a 6-in borehole with wellbore storage and wellbore damage was simulated. Oscillatory pumping was approximated by 1-second stress periods during a 24-second cycle, where the peak flow rate was 0.92 gpm (5.8E-05 m³/s). Twelve cycles were simulated and water-level changes from the last two cycles were presented (Figure 1KJH). The pumping well was simulated with the first column as 10 ft of cells with very high hydraulic conductivities, a specific yield of 1, and specific storages of 0. Gravel pack of the annular space between well screen and formation was simulated with the second column and extended from 5 to 10

ft below the water table. Hydraulic conductivity of the gravel pack (Kann) ranged from 0.1 ft/d for significant wellbore damage to 100 ft/d for a fully developed well.

Horizontal hydraulic conductivities between 1 and 300 ft/d could reasonably be estimated with oscillatory pumping. Water-level changes for Kx = 3 ft/d were best suited to analysis with the initial displacement largely dissipated and wellbore damage minimally affecting estimates (Figure 1KJH). Water-level changes for Kx = 100 ft/d were sensitive to wellbore damage, which causes Kx to be underestimated. Measurements of water-level changes approach inherent noise in a pumped well as Kx exceeds 300 ft/d. Slug tests are limited equally by wellbore damage and an upper threshold for estimating Kx. Water-level changes for Kx = 0.1 ft/d show a greater sensitivity to dissipating the initial displacement than amplitude of water-level changes. Amplitude was 0.31 ft for both Kx of 0.1 and 3 ft/d (Figure 1KJH), which suggests the oscillatory signal is insensitive to Kx < 1 ft/d. Slug tests are more appropriate for small hydraulic conductivities because recovery time is the limiting factor. About 100 minutes is the 90-% recovery time for a slug test in the example with Kx = 0.1 ft/d. Reasonable variations in well construction and peak pumping rates will not alter these conclusions.

Response: Our response to the comments is as follows:

(1) We presented not only transient solution considering the initial condition of static groundwater for OPT but also pseudo-steady sate solution assuming head fluctuation as a simple harmonic motion (SHM). The former solution is applicable to the case of $K_x$ = 0.1 ft/d with significant initial displacement as discussed in section 3.1 of the manuscript. We believe the comment *"Amplitude was 0.31 ft for both Kx of 0.1 and 3 ft/d (Figure 1KJH), which suggests the oscillatory signal is insensitive to Kx < 1 ft/d."* is true for pseudo-steady sate models rather than transient models.

(2) A key advantage of OPT as mentioned above is its ability to extract the signal from a variety of different types of noise, even when the signal is much smaller than the level of noise, provided the duration of OPT is long enough (Bakhos et al., 2014). Despite such a finding, we agree to the difficult identification of water-level fluctuation for a large hydraulic conductivity because the practical duration of OPT is always limited.

Slug tests investigate significantly larger volumes of aquifer than oscillatory pumping (Figure 2KJH). An initial displacement of 3 ft (1.1 gallons in a 3-in well) simulated a slug test where Kx = 3 ft/d. Water levels were displaced at least 0.02 ft through the thickness of aquifer and about 50 ft away from the slug test. Maximum displacements greater than 0.01 ft from oscillatory pumping mapped where amplitudes exceeded 0.02 ft and were comparable to maximum extents for slug tests. Oscillatory pumping influenced about 2 percent of the aquifer that was influenced by a typical slug test (Figure 2KJH).

Response: We agree that for this case wellbore storage effect diminishes the radius of influence (RI) from the pumping well for OPT and the slug test induces a wider RI due to no wellbore storage effect. However, the OPPT is a good alternative because of using piston pumping to prevent the effect. If this case adopts the OPPT with 50-second cycle, the RI at which the amplitude is 0.02 ft extends to about 50 ft predicted by the present solution. The associated text is revised to accommodate the comment.

The need for oscillatory pumping as an aquifer-testing method seems limited, which has greatly dampened my enthusiasm for reviewing anymore articles about oscillatory pumping.
Sincerely, Keith J Halford

Response: The OPT has its advantages over other tests as mentioned above. The OPT was adopted for field experiments with success in the past. It may be a feasible alternative in the case that the field test condition is not favorable to other tests.

**References**

Bakhos, T., Cardiff, M., Barrash, W., and Kitanidis, P. K.: Data processing for oscillatory pumping tests, J. Hydrol., 511, 310–319, 2014.

Black, J. H., and Kipp, K. L.: Determination of Hydrogeological Parameters Using Sinusoidal Pressure Tests - a Theoretical Appraisal, Water Resour. Res., 17(3), 686–692, 1981.

Cardiff, M., Bakhos, T., Kitanidis, P. K., and Barrash, W.: Aquifer heterogeneity characterization with oscillatory pumping: Sensitivity analysis and imaging potential, Water Resour. Res., 49(9), 5395–5410, 2013.

Dagan, G. and Rabinovich, A.: Oscillatory pumping wells in phreatic, compressible, and homogeneous aquifers, Water Resour. Res., 50(8), 7058–7066, 2014.

Rabinovich, A., Barrash, W., Cardiff, M., Hochstetler, D., Bakhos, T., Dagan, G., and Kitanidis, P. K.: Frequency dependent hydraulic properties estimated from oscillatory pumping tests in an unconfined aquifer, J. Hydrol., 531, 2–16, 2015.

Spane, F. A., and Mackley, R. D., Removal of river-stage fluctuations from well response using multiple regression, Ground Water, 49, 794–807, 2011.

**Figures**

[Figure]

**Figure 1KJH.—Water-level changes in an oscillatory pumping well for horizontal hydraulic conductivities (Kx) of 0.1, 3, and 100 ft/d and annular hydraulic conductivities (Kann) of 0.1, 3, and 100 ft/d.**

[Figure]

**Figure 2KJH.—Maximum extent of water-level displacement from a slug test with 3 ft (1.1 gal) of initial displacement and oscillatory pumping with a peak rate of 0.92 gpm, where Kx = 3ft/d and Kann = 100 ft/d.**

---

## Author Comment (AC2) · 10 Jun 2018

**Reply to Reviewer 2**

Review of: "Analysis of Groundwater Response to Oscillatory Pumping Test in Unconfined Aquifers: Consider the Effects of Initial Condition and Wellbore Storage" By Ching-Sheng Huang, Ya-Hsin Tsai, Hun-Der Yeh, and Tao Yang

Review by: Michael Cardiff

This paper is not acceptable in its present format for at least a few reasons. My primary reason is this: the authors have claimed to have used data from oscillatory pumping tests (data collected at the Boise Hydrogeophysical Research Site (BHRS), by myself and colleagues). Looking at the data they claim to fit, I can guarantee it is not raw data from any of the tests we collected. As far as I am aware, the authors of this paper did not contact any of the primary collectors of this data in an effort to understand it, nor did they apply an analysis strategy that is appropriate. Publishing data that is suspect under the name of the workers from the BHRS (and using a flawed analysis to do so) negatively impacts those who have worked so hard to collect the high quality data available from this site.

Response: One of the authors, Ya-Hsin Tsai (email: yahsinamlaiy4433@gmail.com), did ask four authors of Rabinovich et al. (2015) including the reviewer by email for the BHRS data. The first email was sent on 2016/10/26 to Avinoam Rabinovich (avinoam_r@yahoo.com) who suggested us asking Warren Barrash (wbarrash@cgiss.boisestate.edu). We then sent emails to Warren Barrash on the same date, Michael Cardiff (cardiff@wisc.edu) on 2016/12/09, and Tania Bakhos (tbakhos@bcamath.org) on 2016/12/09, but unfortunately did not receive any response from them. Until now we still keep the letters of those emails.

The data presented in Fig. 8 of our paper was read from Fig. 4(a) of Rabinovich et al. (2015) using the Grapher digitize function. We will appreciate it if the raw data can be provided from one of the authors and let us redo the analysis.

The current paper claims to develop a novel method for analyzing fully-penetrating oscillatory tests in

which wellbore storage and the water table are taken into account. Applying this model to our data from the BHRS is completely nonsensical because: 1) While the wellbore we pumped from was indeed fully penetrating, the wellbore was packed off above and below our "oscillation zone", meaning that only a 1m interval (partially penetrating) zone served as the pumping interval. This does not fit with the model that has been developed in this paper; also 2) There is no need to consider wellbore storage for the tests performed at Boise because we used a piston to generate the signal within the well (i.e., the oscillating zone was under confined conditions, and we forced water into / out of the formation via piston). For both these reasons, the model the authors have developed is inappropriate for analyzing our data. The authors may have found this out earlier had they bothered to contact any of the field workers who spent such time and effort collecting this data.

Response: Thanks for the comment on the use of partially packed screen. The inner boundary condition describing flux across the screen of a fully penetrating well is therefore changed to

$$2\pi r_w K_r l \frac{\partial h}{\partial r} - \pi r_c^2 \frac{\partial h}{\partial t} = \begin{cases} Q\sin(\omega t) & \text{for } z_l \le z \le z_u \\ 0 & \text{outside screen interval} \end{cases} \quad \text{at } r = r_w \tag{R1}$$

where $h$ is hydraulic head, $r$ is radial distance from the centerline of the pumping well, $z$ is elevation, $t$ is time, $K_r$ is radial hydraulic conductivity, $r_c$ and $r_w$ are respectively inner and outer radiuses of the well, $z_l$ and $z_u$ are respectively lower and upper elevations of screen interval, and $l = z_u - z_l$ is screen length. Eq. (R1) considers the well radius and avoids using the assumption of infinitesimal radius as adopted in some articles (e.g., Black and Kipp, 1981; Rasmussen et al., 2003; Dagan and Rabinovich, 2014). A new solution applicable to a well with partial penetration based on Eq. (R1) is presented in the revised manuscript. This new solution can reduce to the special case of no wellbore effect if letting $r_c = 0$. We add a new section in the revised manuscript to present the special case and others for different scenarios.

We agree that the wellbore storage effect might be very small and negligible in the oscillatory piston pumping test (OPPT). The aim of our work, however, is to present a general analytical solution for the cases with various types of oscillatory pumping test (OPT) including the OPPT. Curve fitting

to the data will be reconducted using the present solution with and without considering the wellbore storage effect. The associated text will be rewritten to accommodate all the comments.

With regards to the scientific merit / value of the model itself – I also question whether this model is necessary or useful, and whether it is being considered for reasonable ranges of the given parameters. Consider Figure 5 – Figure 5(b) shows somewhat of a difference from the Dagan and Rabinovich solution at a distance of $\bar{r}$ = 16. Given the non-dimensionalization used, this means it is at a distance of 16 well radii. A standard well radius is about 5 cm, meaning that this effect is being observed only at a distance of less than 0.8m from the pumping location. I have never in my life seen wells spaced 80cm apart. A very big well might be 20 cm, for which the effect would apparently decay after only 3.2m.

Response: The difference in the hydraulic heads predicted by our solution and the Dagan and Rabinovich solution in Fig. 5(b) is arisen from the assumption of infinitesimal radius. The difference is negligible if the distance $r$ from the pumping well to the observation well exceeds 16 times radius $r_w$ of the pumping well. (i.e., $r/r_w \geq 16$). We believe the finding of $r/r_w \geq 16$ can serve as a useful criterion for those who can be aware of the deviation in the results when applying finite difference scheme that treats a pumping well as a nodal point with infinitesimal radius. On the other hand, large-diameter wells with radius ranging from 0.5 m to 2 m are commonly installed in many countries to meet a large demand for domestic and irrigation water uses (Yeh and Chang, 2013). With considering a large-diameter well as a pumping well, the well distance should exceed 8 m for $r_w$ = 0.5 m and 32 m for the extreme case of $r_w$ = 2 m.

The authors seem to have chosen parameters that are unrealistic for most aquifers. For example, they use a specific yield value of $S_y$ = 0.1. Specific yield values in aquifer pumping tests have almost never been measured to be this high (due to delayed drainage), and in the special case of oscillator tests where saturation changes rapidly, it is unlikely even partial drainage will occur. Similarly, many of the

other choices in the plots are suspect. Looking at the definition of $\alpha = \dfrac{r_c^2}{2r_w^2 S_s b}$, for example, I find it hard to understand why the authors have focused on cases such as $\alpha = 1$ and below in Figure 5. Given that Ss is generally in the range of $10^{-5} m^{-1}$ to $10^{-6} m^{-1}$ for any natural material, and that reasonable aquifers may be 10-1000m thick, can one imagine any realistic solutions where $\alpha < 1$?

Response: Thanks for the comment on the magnitude of specific yield $S_y$. Freeze and Cherry (1979, p. 61) mentioned that the usual range of $S_y$ is $0.01 - 0.3$. Todd and May (2005, p. 51) provided a table showing representative specific yields ranging from 0.06 to 0.44 for various geologic materials. With $S_y = 10^{-4}$, the program for the present solution is rerun; all the figures are replotted; some discussions are revised. Regarding the magnitude of $\alpha$, exploring how wellbore storage affects head fluctuation due to OPT was recommended by Prof. John L. Wilson, a 2006 AGU fellow, at 2014 AGU Fall Meeting. The finding is that wellbore storage effect can be ignored for $\alpha \leq 10^{-1}$ and is significant for the practical range of $\alpha > 1$. We believe the finding herein is useful for both OPPT without the wellbore storage effect and other types of OPT which might be subject to significant wellbore storage effect. The associated text is rewritten to accommodate the viewpoint above.

It is also notable that in Figure 8, the confined solution appears to fit the data perfectly well (using the same K and Ss parameters as the unconfined solution, if I am reading correctly) almost exactly as well as the more complex model. This would indicate to me that the details considered in this more complex model matter not one bit, and the water table can simply be considered as a no-flux boundary practically in these tests.

Response: The case of confined flow for curve fitting in Fig. 8 has been removed for avoiding confusion that the water table can be regarded as no-flow boundary. The associated text will be rewritten to accommodate all the comments.

Similarly, Figure 5(a) represents head at the edge of the wellbore itself, which is unlikely to be used in

real field scenarios because measurements at the pumping location are subject to numerous nuisance factors (for example, wellbore "skin", non-darcian flow conditions near the wellbore, inertial effects, etc. So I see no practical reason to consider the variability in this result.

Response: We would like to mention that the radius of the pumping well is not negligible in spite of no wellbore storage effect as indicated in the figure when the head is measured at the rim of the well. Literature review reveals there are many scenarios where time-depending head data were measured at the rim of a pumping well (e.g., Pacheco, 2002; Mohamed and Rushton, 2006; Rabinovich et al., 2015).

While it is mathematically interesting to derive new PDE solutions, I fail to see the practical application of these much more complex solutions, given that they are still invoking many assumptions / approximations. For example, the authors do not deal with the fact that they are using only an approximation for the water table response (the linearized free surface condition of Neumann), and that realistically oscillatory tests are likely to be subject to delayed drainage and differing yields as a function of frequency. For all of these reasons I cannot recommend that this paper be published.

Response: The manuscript is largely revised for accommodating the comments above. A more general analytical solution of hydraulic head for OPT is derived when the original inner boundary condition is replaced by Eq. (R1) and the linearized free surface equation is replaced by

$$K_z \frac{\partial h}{\partial z} = -\varepsilon\, S_y \int_0^t \frac{\partial h}{\partial t'} \exp(-\varepsilon(t - t'))\, \mathrm{d}t' \quad \text{at} \quad z = b \tag{R2}$$

where $K_z$ is vertical aquifer hydraulic conductivity, $b$ is aquifer thickness, $\varepsilon$ is an empirical constant, and the term on the right-hand side accounts for the effect of delayed gravity drainage (Moench, 1995). New results are discussed and new conclusions are drawn based on the new solution. To conclude our new results, our solution has the following advantages:

(1) The new transient or pseudo-steady sate solution can reduce to seven special cases classified according to aquifer type (i.e., confined or unconfined), well penetration (i.e., full or partial), and with or without wellbore storage effect.

(2) It can be a handy tool to design the OPT or to estimate hydraulic parameters when coupled with an optimization approach because of its simplicity and considerations of well radius, wellbore storage and delayed yield.

(3) It can find a transient head directly at any time $t$. With a small time step for obtaining accurate results, numerical methods such as finite difference and finite element methods should compute the heads at each time step until reaching the end of timeframe, leading to a large amount of computing time.

(4) It allows us to find a head solution at any specific location ($r$, $z$) without going through the entire marching process of finding heads at all nodal points, as one does in finding the numerical solutions using the finite difference or finite element method.

(5) It is stable, efficient, and easier to implement in solution evaluations.

**References**

Black, J. H., and Kipp, K. L.: Determination of Hydrogeological Parameters Using Sinusoidal Pressure Tests - a Theoretical Appraisal, Water Resour. Res., 17(3), 686–692, 1981.

Dagan, G. and Rabinovich, A.: Oscillatory pumping wells in phreatic, compressible, and homogeneous aquifers, Water Resour. Res., 50(8), 7058–7066, 2014.

Freeze, R. A., and Cherry, J. A.: Groundwater, Prentice-Hall, New Jersey, 1979.

Moench, A. F.: Combining the Neuman and Boulton models for flow to a well in an unconfined aquifer, Ground Water, 33(3), 378–384, 1995.

Mohamed, A., Rushton, K.: Horizontal wells in shallow aquifers: field experiment and numerical model, J. Hydrol., 329, 98–109, 2006.

Pacheco, F. A. L.: Response to pumping of wells in sloping fault zone aquifers, J. Hydrol., 259(1-4), 116–135, 2002.

Rabinovich, A., Barrash, W., Cardiff, M., Hochstetler, D., Bakhos, T., Dagan, G., and Kitanidis, P. K.: Frequency dependent hydraulic properties estimated from oscillatory pumping tests in an

unconfined aquifer, J. Hydrol., 531, 2–16, 2015.

Rasmussen, T. C., Haborak, K. G., and Young, M. H.: Estimating aquifer hydraulic properties using sinusoidal pumping at the Savannah River site, South Carolina, USA, Hydrogeol. J., 11(4), 466–482, 2003.

Todd, D. K. and Mays, L. W.: Groundwater Hydrology, 3rd ed., John Wiley & Sons, NY, 2005.

Yeh, H. D. and Chang, Y. C.: Recent advances in modeling of well hydraulics, Adv. Water Resour., 51, 27–51, 2013.

---

## Short Comment (SC1) · 20 Jun 2018

This is a comment on the Reply to Reviewer 2 (hess-2018-199-AC2-supplement.pdf).

I am familiar with the subject and the data which the authors have misrepresented and misused in the original manuscript, as noted below. The data (in Fig. 8 of the ms and in the text of the ms) were misrepresented as being actual field data from the experiment at the BHRS. The authors admit this in their AC-2 supplement where they say they digitized data from Fig. 4(a) of Rabinovich et al. 2015. *** But the data do not

look at all similar (data are too smooth and they unrealistically match ideal oscillations, and they are not at the same relative amplitude and phase positions), so the authors' statement notwithstanding, the authors presented and analyzed a very suspect digitizing operation and/or very suspect transcription to the figure without quality control as expected for scientific research - and then analyzed and interpreted the "data" with the false attribution to our OHT work at the BHRS. ***

Furthermore, as also noted by Dr. Cardiff, the authors missed the essential details that the experiment was conducted in wells subdivided by straddle packers, i.e., not conducted in open fully penetrating wells. And the actual oscillatory pumping configuration does not involve any wellbore storage. *** That is, their premise for using the data is unfounded. ***

The email I received in October 2016 was very perfunctory and did not indicate an understanding of the need to consider context and metadata in order to use field data properly. I do not feel obligated to respond to such inquiries that essentially say: "send me the data," and I do not have the time to engage in discussion to find out what the person wants or needs, and then (as may be necessary if people are unfamiliar with field experiments or field data) to guide and review to be sure the data are treated and/or used properly. I am speaking from experience on this.

So now, given the above experience with the authors, the Reply to Reviewer 2 states they want to do a reanalysis and modify their analytical model to include partially penetrating wells and thereby match the OHT field pumping configuration. But they go on to say "Curve fitting to the data will be conducted using the present solution with and without considering the wellbore storage effect."

*** But this is still an inappropriate use of the data they are requesting. It doesn't make sense to analyze data with an inappropriate model (which the authors explicitly acknowledge is their intention in their reply) in order to compare the results with modeling using data appropriate for the model. ***
If the analysis is conducted correctly, it will be a foregone conclusion that the results will differ and the real data from the BHRS will look bad by comparison. This is not an appropriate use of the OHT data we collected, and there is no sound reason to put the data in the literature in a confusing context. Anyone who is competent in collecting and modeling/analyzing OHT field data would know not to use a wellbore storage model with data such as the BHRS data.

I strongly recommend that the ms by the authors not be published with BHRS OHT data. I recommend that the authors find other data that are appropriate for their analysis, or better yet, collect their own data.

Furthermore, on another issue, the authors have missed the point about small specific yield values for short duration tests that have been reported repeatedly in the literature (see citations at the end of these comments to Neuman 1975; Moench 1994; Chen and Ayers 1998; Barrash et al. 2006 - all from or cited in Barrash et al. 2006 on the BHRS aquifer - which is cited in Rabinovich et al. 2015). The authors cite general text books that give specific yield values from drainage over considerably longer periods of time than those of a short-duration pumping test - i.e., inappropriate for the subject of the paper and missing the relevant and well-documented parameter range.

Barrash, W., Clemo, T., Fox, J.J., and Johnson, T.C., 2006, Field, laboratory, and modeling investigation of the skin effect at wells with slotted casing: Journal of Hydrology, v. 326, no. 1-4, p. 181-198, doi:10.1016/j.jhydrol.2005.10.029.

Chen, X. and Ayers, J.F., 1998, Aquifer properties determined from two analytical solutions. Ground Water, 36(5), 783-791.

Moench, A.F., 1994, Specific yield as determined by type-curve analysis of aquifer-test data. Ground Water, 32(6): 949-957.

Neuman, S.P., 1975, Analysis of pumping test data from anisotropic unconfined aquifers considering delayed gravity response. Water Resources Research 11(2), 329-

342.

Interactive
comment

---

## Author Comment (AC3) · 29 Jun 2018

Reviewed by W. Barrash

This is a comment on the Reply to Reviewer 2 (hess-2018-199-AC2-supplement.pdf). I am familiar with the subject and the data which the authors have misrepresented and misused in the original manuscript, as noted below. The data (in Fig. 8 of the ms and in the text of the ms) were misrepresented as being actual field data from the experiment at the BHRS. The authors admit this in their AC-2 supplement where they say they digitized data from Fig. 4(a) of Rabinovich et al. 2015. *** But the data do not look at all similar (data are too smooth and they unrealistically match ideal oscillations, and they are not at the same relative amplitude and phase positions), so the authors' statement notwithstanding, the authors presented and analyzed a very suspect digitizing operation and/or very suspect transcription to the figure without quality control as expected for scientific research - and then analyzed and interpreted the "data" with the false attribution to our OHT work at the BHRS. ***

Response: Rabinovich et al. (2015) mentioned that "In this work we focus our attention on the first dominant frequency, using only a single harmonic to represent the signal and thus simplifying computations in the comparison with the semi-analytical solution." in the third paragraph of section 3.2. The field test results for the hydraulic head as a function of time shown in Fig. 4(a) of their paper is given at the end of this reply as panel (a) of Fig. R1. In addition, they also mentioned that "First, two initial periods of data were removed in order to avoid transient flow associated with the onset of pumping, thus assuring steady periodic flow." in the second paragraph of section 3.2. We therefore digitized the portion of their f1(FFT) data in panel (b) of Fig. R1 because the head fluctuation can be regarded as a simple harmonic motion (i.e., steady periodic flow). It is apparent to see that our digitized data in Fig. R1 are very close to their f1 data in either the amplitude or the phase. Our data may have

some reading errors but their influence on the parameter estimation is negligible from the least-squares sense. We think there might be a misunderstanding in their comments as follows "the authors have misrepresented and misused in the original manuscript" and "they are not at the same relative amplitude and phase positions".

Furthermore, as also noted by Dr. Cardiff, the authors missed the essential details that the experiment was conducted in wells subdivided by straddle packers, i.e., not conducted in open fully penetrating wells. And the actual oscillatory pumping configuration does not involve any wellbore storage. ***
That is, their premise for using the data is unfounded. ***
Response: As we stated in the Reply to Reviewer 2, the inner boundary condition in the previous manuscript describing flux across the screen of a fully penetrating well is changed to a new one that is applicable to a partially penetrating well as below

$$2\pi r_w K_r l \frac{\partial h}{\partial r} - \pi r_c^2 \frac{\partial h}{\partial t} = \begin{cases} Q\sin(\omega t) & \text{for } z_l \leq z \leq z_u \\ 0 & \text{outside screen interval} \end{cases} \quad \text{at} \quad r = r_w \tag{R1}$$

where $h$ is hydraulic head, $r$ is radial distance from the centerline of the pumping well, $z$ is elevation, $t$ is time, $K_r$ is radial hydraulic conductivity, $r_c$ and $r_w$ are respectively inner and outer radiuses of the well, $z_l$ and $z_u$ are respectively lower and upper elevations of screen interval, and $l = z_u - z_l$ is screen length. A new analytical solution based on Eq. (R1) is presented in the revised manuscript. It can reduce to a special case of no wellbore effect when letting $r_c = 0$. With this new solution, we will redo curve fitting in Fig. 8 of our paper. Therefore, we sincerely hope that the reviewers can kindly provide us the full time series and f1(FFT) data shown in their figure (i.e., in Fig. 4(a) of Rabinovich et al. (2015)).

The email I received in October 2016 was very perfunctory and did not indicate an understanding of the need to consider context and metadata in order to use field data properly. I do not feel obligated to respond to such inquiries that essentially say: "send me the data," and I do not have the time to engage

in discussion to find out what the person wants or needs, and then (as may be necessary if people are unfamiliar with field experiments or field data) to guide and review to be sure the data are treated and/or used properly. I am speaking from experience on this.

Response: Our email explicitly stated the need of time-varying hydraulic head data in Figure 4 of Rabinovich et al. (2015) for parameter estimation. The original letter by the second author to the reviewer Dr. W. Barrash is given below: "I'm a graduate student working on a study related to the topic of parameter estimation for oscillatory pumping test data. … In the paper, the hydraulic head data measured at the Boise Site was analyzed and illustrated in Figure 4. Would you please provide the original measured hydraulic head versus time data and the after applying the Fourier analysis data? …"

So now, given the above experience with the authors, the Reply to Reviewer 2 states they want to do a reanalysis and modify their analytical model to include partially penetrating wells and thereby match the OHT field pumping configuration. But they go on to say "Curve fitting to the data will be conducted using the present solution with and without considering the wellbore storage effect."

*** But this is still an inappropriate use of the data they are requesting. It doesn't make sense to analyze data with an inappropriate model (which the authors explicitly acknowledge is their intention in their reply) in order to compare the results with modeling using data appropriate for the model. ***

If the analysis is conducted correctly, it will be a foregone conclusion that the results will differ and the real data from the BHRS will look bad by comparison. This is not an appropriate use of the OHT data we collected, and there is no sound reason to put the data in the literature in a confusing context. Anyone who is competent in collecting and modeling/analyzing OHT field data would know not to use a wellbore storage model with data such as the BHRS data.

I strongly recommend that the ms by the authors not be published with BHRS OHT data. I recommend that the authors find other data that are appropriate for their analysis, or better yet, collect their own data.

Response: Rabinovich et al. (2015) applied the analytical solution of Dagan and Rabinovich (2014) to analyze data obtained from the oscillatory piston pumping test conducted at the Boise Hydrogeophysical Research Site (BHRS). In spite of no wellbore storage effect (i.e., $r_c = 0$), our new solution in the revised manuscript has two superiorities over their solution. One is to consider the finite radius of an oscillatory pumping well expressed as Eq. (R1), and the other is to include the delayed yield effect denoted as

$$K_z \frac{\partial h}{\partial z} = -\varepsilon S_y \int_0^t \frac{\partial h}{\partial t'} \exp(-\varepsilon(t - t')) \, \mathrm{d}t' \quad \text{at} \quad z = b \qquad (R2)$$

where $K_z$ is vertical aquifer hydraulic conductivity, $b$ is aquifer thickness, $\varepsilon$ is an empirical constant, and the term on the right-hand side accounts for the effect of delayed gravity drainage (Moench, 1995). Since their solution is applicable to the BHRS data, our new solution should be applicable as well. The comment "It doesn't make sense to analyze data with an inappropriate model" is his prejudice.

Furthermore, on another issue, the authors have missed the point about small specific yield values for short duration tests that have been reported repeatedly in the literature (see citations at the end of these comments to Neuman 1975; Moench 1994; Chen and Ayers 1998; Barrash et al. 2006 - all from or cited in Barrash et al. 2006 on the BHRS aquifer - which is cited in Rabinovich et al. 2015). The authors cite general text books that give specific yield values from drainage over considerably longer periods of time than those of a short-duration pumping test - i.e., inappropriate for the subject of the paper and missing the relevant and well-documented parameter range.

Response: We didn't miss the point about the reasonable values of $S_y$. Reviewer 2 gave the following comments: "The authors seem to have chosen parameters that are unrealistic for most aquifers. For example, they use a specific yield value of $S_y = 0.1$." We think the value of $S_y = 0.1$ is reasonable for most aquifers as clearly explained in our response to his comment. Huang and Yeh (2007) proposed an approach for on-line aquifer parameter estimation based on the sensitivity analysis. In their paper, three sets of pumping drawdown data were analyzed. Two sets of them, data sets 1 and 2, are synthetic

with assuming $S_y = 0.1$ while the third set is field pumping data from Cape Cod site (Moench et al., 2000). The well F507-080 was pumped with an average rate 1.21 m$^3$/min for 72 hours and the data were observed at well F505-032. The time-drawdown data set 1 was generated by Neuman's model (1974) for a pumping period of 1 to 176,360 seconds (49 hours) while the data set 2 was generated by Moench's model (1997) for a pumping period of 0.6 to 600,000 seconds (166.7 hours). The time-drawdown data and the related normalized sensitivities are plotted in Fig. R2a (Fig. 2 in Huang and Yeh (2007)) for data set 1 and in Fig. R2b (Fig. 3 in Huang and Yeh (2007)) for data set 2. (Note that the figures mentioned herein are shown at the end of this reply.) This figure clearly indicates the drawdown is very insensitive to the change in $S_y$ in the early period of 1 to 100 seconds. Fig. R3 shows the curve of estimated $S_y$ versus time dramatically fluctuates in the early period and converges to the value of 0.1 after about 80 seconds for data set 1 in panel (a) (Fig. 5 in Huang and Yeh (2007)) and 125 seconds for data set 2 in panel (b) (Fig. 6 in Huang and Yeh (2007)). Fig. R4 (Fig. 7 in Huang and Yeh (2007)) demonstrates the estimated $S_y$ versus time when analyzing the field pumping data from Cape Cod site. The estimated $S_y$ keeps at a value of 0.3 for the first 20 minutes, then goes down rapidly and reaches a minimal at 100 minutes. On the basis of those data analyses, it is obvious that the estimation of $S_y$ using early time (or short duration) data is not reliable.

We realize that the estimated $S_y$ for unconfined aquifers from a constant rate pumping test ranges from 0.01 to 0.3, but from an oscillatory pumping test may be very low. The present solution will be rerun with default value of $S_y = 10^{-4}$ for the analyses given in the Results and Discussion section.

**References**

Barrash, W., Clemo, T., Fox, J.J., and Johnson, T.C.: Field, laboratory, and modeling investigation of the skin effect at wells with slotted casing, Journal of Hydrology, 326(1-4), 181-198, 2006.

Chen, X. and Ayers, J.F.: Aquifer properties determined from two analytical solutions, Ground Water, 36(5), 783-791, 1998.

Huang, Y.C., Yeh, H.D.: The use of sensitivity analysis in on-line aquifer parameter estimation, Journal

of Hydrology, 335(3-4), 406-418, 2007.

Moench, A.F.: Specific yield as determined by type-curve analysis of aquifer-test data, Ground Water, 32(6), 949-957, 1994.

Moench, A.F.: Combining the Neuman and Boulton models for flow to a well in an unconfined aquifer, Ground Water, 33(3), 378–384, 1995.

Moench, A.F.: Flow to a well of finite diameter in a homogenous anisotropic water table aquifer, Water Resources Research, 33, 1397- 1407, 1997.

Moench, A.F., Garabedian S.P., and LeBlanc D.L.: Estimation of Hydraulic Parameters from an Unconfined Aquifer Test Conducted in a Glacial Outwash Deposit, Cape Cod, Massachusetts. USGS Open-File Report: 00-485, 2000.

Neuman, S.P.: Effects of partial penetration on flow in unconfined aquifers considering delayed aquifer response. Water Resources Research, 10, 303-312, 1974.

Neuman, S.P.: Analysis of pumping test data from anisotropic unconfined aquifers considering delayed gravity response, Water Resources Research, 11(2), 329-342, 1975.

Rabinovich, A., Barrash, W., Cardiff, M., Hochstetler, D., Bakhos, T., Dagan, G., and Kitanidis, P. K.: Frequency dependent hydraulic properties estimated from oscillatory pumping tests in an unconfined aquifer, Journal of Hydrology, 531, 2–16, 2015.

**Figures**

[Figure]

**Figure R1.** Field test results for hydraulic head as a function of time and our digitized data from f1(FFT). Panel (a) is adopted from Fig. 4(a) of Rabinovich et al. (2015). Panel (b) shows the agreement on our digitized data and their f1(FFT) data

[Figure]

**Figure R2.** The time-drawdown data and the normalized sensitivities of the unconfined aquifer parameters (a) Neuman's model and (b) Moench's model (from Huang and Yeh (2007))

[Figure]

**Figure R3.** The estimated $S_y$ versus time using (a) data set 1 and (b) data set 2 (from Huang and Yeh (2007))

[Figure]

**Figure R4.** The estimated $S_y$ versus time using field pumping data from Cape Cod site (from Huang

and Yeh (2007))